# A bacterial regulatory uORF senses multiple classes of ribosome-targeting antibiotics

Gabriele Baniulyte[1], Joseph T Wade[1,2,3]*

[1]Department of Biomedical Sciences, School of Public Health, University at Albany, SUNY, Rensselaer, United States; [2]Wadsworth Center, New York State Department of Health, Albany, United States; [3]RNA Institute, University at Albany, SUNY, Albany, United States

**\*For correspondence:**
joseph.wade@health.ny.gov

**Competing interest:** The authors declare that no competing interests exist.

## eLife Assessment

In this **important** study, Baniulyte and Wade provide **convincing** evidence that translation of a short ORF denoted *toiL* positioned upstream of the *topAI-yjhQP* operon is responsive to different ribosome-targeting antibiotics, consequently controlling translation of the TopAI toxin as well as Rho-dependent transcription termination. Strengths of the study include combining a genetic screen to identify 23S rRNA mutations that affect *topA1* expression and a creative approach to map the different locations of ribosome stalling within *toiL* induced by different antibiotics, with ribosome profiling and RNA structure probing by SHAPE to examine consequences of different antibiotics on *toiL*-mediated regulation. The work leaves unanswered how bacteria benefit by activating expression of the genes using the proposed strategy and the mechanism underlying ToiL's sensing of structurally distinct antibiotics.

**Abstract** Expression of many bacterial genes is regulated by *cis-* and *trans*-acting elements in their 5′ upstream regions (URs). *Cis*-acting regulatory elements in URs include upstream ORFs (uORFs), short ORFs that sense translation stress that manifests as ribosomes stalling at specific codons within the uORF. Here, we show that the transcript encoding the *Escherichia coli* TopAI-YjhQ toxin–antitoxin system is regulated by a uORF that we name 'toiL'. We propose that in the absence of translation stress, a secondary structure in the UR represses translation of the *topAI* transcript by occluding the ribosome-binding site. Translation repression of *topAI* leads to premature Rho-dependent transcription termination within the *topAI* ORF. At least five different classes of ribosome-targeting antibiotics relieve repression of *topAI*. Our data suggest that these antibiotics function by stalling ribosomes at different positions within *toiL*, thereby altering the RNA secondary structure around the *topAI* ribosome-binding site. Thus, *toiL* is a multipurpose uORF that can respond to a wide variety of translation stresses.

## Introduction

Expression of many bacterial genes is regulated by elements in their upstream regions (URs) (*Adams et al., 2021*). UR RNA can contain binding sites for *trans*-acting factors such as regulatory RNAs or RNA-binding proteins. UR RNAs can also include *cis*-acting elements, such as sequences that form secondary structures, and short ORFs (sORFs) known as upstream ORFs (uORFs). Regulatory elements in URs can modulate transcription termination within the UR, RNase accessibility within the UR, or translation initiation of the downstream gene.

uORFs are typically <50 codons in length, and sense translational stress to regulate expression of the downstream gene. Environmental perturbations promote ribosome stalling within uORFs, often in a sequence-specific manner *Ramu et al., 2009*; stalled ribosomes alter the UR RNA secondary structure, modulating the formation of transcription terminators, the accessibility of the ribosome binding site for the downstream gene, or loading sites for the Rho transcription termination factor. Ribosome stalling within uORFs can be mediated by limiting concentrations of charged tRNAs (*Landick and Yanofsky, 1987*) or by small molecules that bind the translating ribosomes, including amino acids and antibiotics (reviewed by *Seip and Innis, 2016*).

Translation repression mediated by regulatory elements in URs can lead to an additional level of repression at the level of transcription termination (*Adams et al., 2021*; *Baniulyte et al., 2017*; *Bastet et al., 2017*; *Bossi et al., 2012*; *Yakhnin et al., 2001*). This occurs because Rho-dependent transcription termination is inhibited by translation (*Adhya and Gottesman, 1978*; *Richardson, 2002*). Reduced translation mediated by regulatory elements in URs allows Rho to load onto Rho utilization (Rut) sequences in the nascent RNA for the ORF that would otherwise be occluded by ribosomes. Ruts are enriched for 'YC' dinucleotides, lack extensive secondary structure, and have a high C:G ratio (*Alifano et al., 1991*; *Nadiras et al., 2018*; *Rivellini et al., 1991*; *Schneider et al., 1993*). The relatively low information content required for a Rut means that many sequences within ORFs can function as a Rut when translation is repressed. Translating ribosomes may also inhibit Rho termination by preventing association of Rho with RNA polymerase (RNAP) and the associated transcription elongation factor NusG (*Burmann et al., 2010*).

Genome-wide studies of transcription termination in *Escherichia coli* and *Mycobacterium tuberculosis* have identified many Rho termination events within ORFs (*Adams et al., 2021*; *Cardinale et al., 2008*; *Dar and Sorek, 2018*; *D'Halluin et al., 2022*; *Peters et al., 2012*; *Peters et al., 2009*), suggesting that modulation of premature Rho termination is a widespread regulatory mechanism. Here, we characterize the mechanism of regulated Rho termination within the *E. coli topAI* gene that is operonic with two additional genes, *yjhQ* and *yjhP* (*Figure 1A*; *Mao et al., 2015*). The *topAI* and *yjhQ* genes encode a type II toxin-antitoxin system, where *topAI* encodes a topoisomerase A inhibitor, and *yjhQ* encodes the cognate antitoxin (*Yamaguchi and Inouye, 2015*). We show that Rho-dependent transcription termination within *topAI* is a consequence of translation repression. We further show that the long 5′ UR of *topAI* encodes a regulatory uORF, *toiL* (*Figure 1A*), that acts as a sensor for translation stress caused by multiple classes of ribosome-targeting antibiotics. We propose that these antibiotics promote ribosome stalling at various sites within *toiL*, and that ribosome stalling anywhere within *toiL* unmasks the ribosome binding site of *topAI*. Thus, *toiL* is a multipurpose regulatory uORF that can sense a wide variety of translational stresses.

## Results

### Translational repression of *topAI* leads to intragenic Rho-dependent transcription termination

Genome-scale analysis of Rho termination identified a putative termination site in the coding region of *topAI* (*Adams et al., 2021*; *Peters et al., 2012*). The *topAI* transcript has an unusually long UR (171 nt) (*Thomason et al., 2015*), suggesting that Rho termination within *topAI* may be regulated by sequences in the UR. To independently assess whether Rho prematurely terminates *topAI* transcription, we constructed a *lacZ* transcriptional reporter fusion that included the *topAI* promoter and UR, and 42 nt of the *topAI* coding region (*Figure 1B*). We measured expression of this reporter fusion in wild-type cells and cells expressing the R66S Rho mutant that is defective for RNA binding and termination (*Baniulyte et al., 2017*; *Martinez et al., 1996*). We observed approximately sevenfold higher expression in *rho* mutant cells than in wild-type cells (*Figure 1C*), consistent with a Rho termination site upstream of position 42 of the *topAI* gene.

The precise length and position of a *rut* is difficult to predict since Rho binds RNA with relatively low specificity; however, Rho is believed to favor pyrimidine-rich, unstructured RNA regions (*Chhakchhuak et al., 2018*; *Nadiras et al., 2018*). The most pyrimidine-rich region of the *topAI* UR is at the very 5′ end. Mutating just four C residues in the Rho-terminated construct of *topAI* (*Figure 1B*) caused a fourfold increase in expression in wild-type cells, but no change in expression in *rho* mutant cells

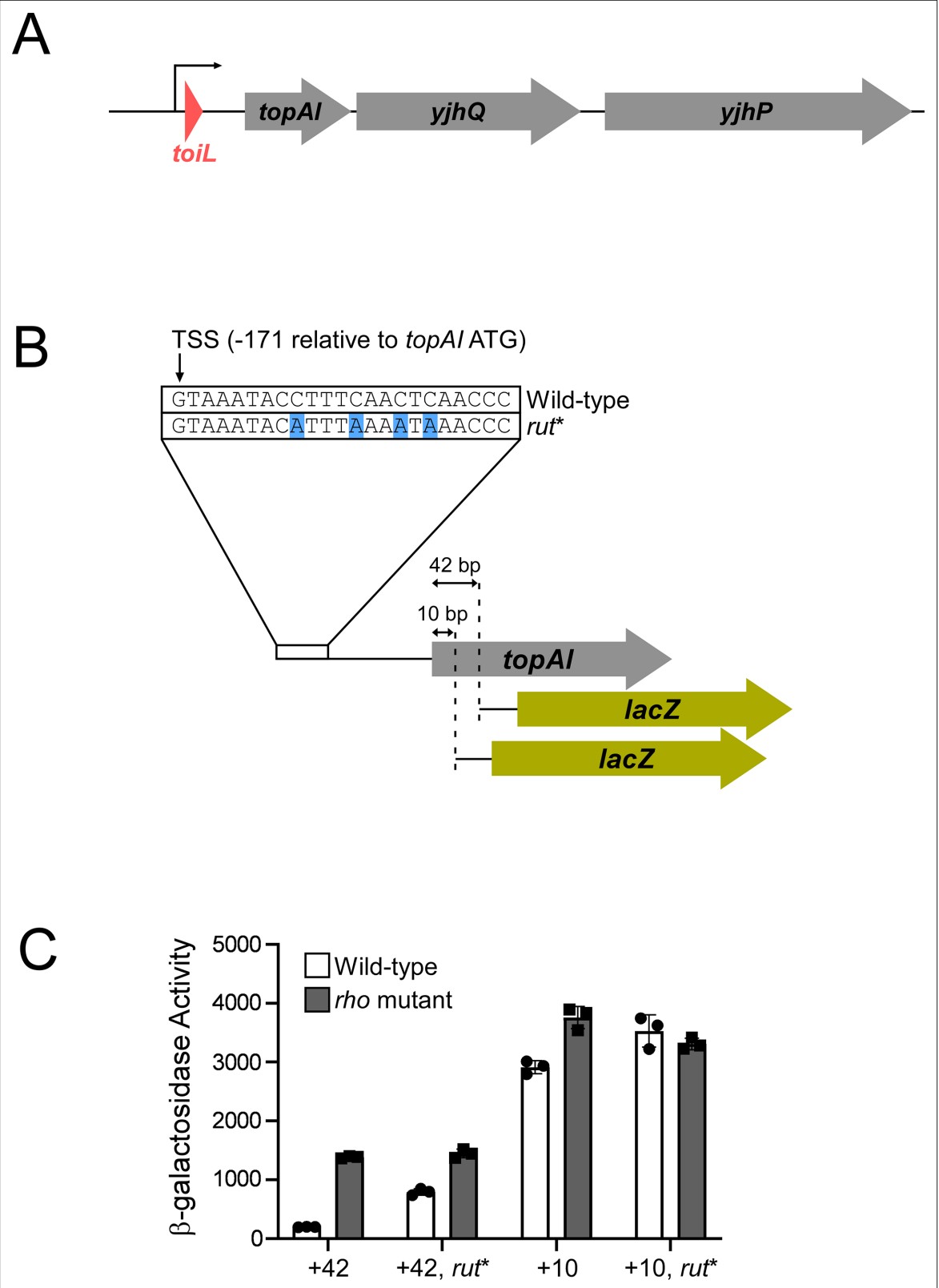

**Figure 1.** Rho-dependent transcription termination within the *topAI* gene. (**A**) Schematic representation of the *E. coli topAI-yjhQ-yjhP* operon, including the *toiL* uORF. (**B**) Schematic representation of *topAI-lacZ* transcriptional reporter fusions, indicating the position of the transcription start site (TSS) with respect to the *topAI* start codon, and the mutation in the putative *rut*. (**C**) β-galactosidase activity of *topAI-lacZ* transcriptional fusions in wild-type (strain AMD054) or *rho* mutant (R66S amino acid substitution in Rho; strain GB004) cells. Reporter fusions with wild-type sequence or mutation of the predicted

Figure 1 continued

rut (rut*) extend to position + 42 or +10 of topAI (wild-type + 42, plasmid pGB217; wild-type + 10, plasmid pGB215; rut* + 42, plasmid pGB306; rut* +10, plasmid pGB305). Error bars represent ±1 standard deviation from the mean (n=3).

(Figure 1C). We conclude that the rut includes the pyrimidine-rich sequence at the very 5' end of the topAI transcript.

To determine the position of the Rho termination site, we measured transcriptional activity of a 'short' topAI::lacZ fusion that includes only the first 10 bp of the topAI ORF (Figure 1B). The effect of mutating rho on the expression of this short reporter fusion was substantially lower than that observed for the longer reporter fusion, with no difference in expression observed for the short reporter fusion containing the mutant rut sequence (Figure 1C). This suggests that most of the transcription termination occurs between nucleotides 10 and 42 of the topAI coding region. We conclude that Rho loads onto the RNA early in the topAI transcript, but does not trigger termination until RNAP is within the ORF, >180 nt further downstream.

Translating ribosomes protect nascent mRNA in protein-coding regions from Rho (de Smit et al., 2009). Given that Rho termination occurs within the topAI ORF, we speculated that topAI is translationally repressed. To test whether topAI is actively translated, we constructed a topAI-lux translational reporter fusion (Figure 2A, Figure 2—figure supplement 1A). We compared expression of this reporter fusion in wild-type and rho mutant cells. Although transcriptional repression of topAI is relieved in the rho mutant background (Figure 1), we observed strong repression of the topAI-lux translation in both wild-type and rho mutant cells (Figure 2—figure supplement 1B). These data suggest that Rho termination within the topAI coding region is a consequence of translational repression.

## topAI repression is relieved under conditions of translation stress

Translation repression often occurs by binding of a trans-acting factor (e.g., protein, sRNA, small molecule) overlapping the ribosome-binding site (Breaker, 2018; Kriner et al., 2016). To identify trans-acting regulators of topAI, we used a genetic selection for spontaneous mutants with increased topAI expression. Briefly, the mutant selection used a topAI-lacZ construct in a ΔlacZ background; overnight cultures were plated on minimal medium with lactose as the only carbon source, only allowing growth of spontaneous mutants with upregulated topAI expression (cis-acting mutants were discarded). We isolated 42 mutants with upregulated topAI expression. Of the mutants, 39 had a mutant rho gene (see 'Materials and methods' for details), suggesting that the screen was saturating. We isolated three additional mutants that each carry single base mutations in one of the seven copies of the 23S rRNA gene, in domain IV, helix 69 (rrlA ΔG1911, rrlC ΔT1917) or domain V, helix 80 (rrlA G2253T) (Supplementary file 1A). Mutations in these regions of 23S rRNA have been suggested to cause nonsense and frameshift readthrough, defects in translation fidelity and ribosome assembly, and reduced binding of ribosome release factors (Feinberg and Joseph, 2006; Gregory et al., 1994; Hirabayashi et al., 2006; Kipper et al., 2011; Liiv et al., 2005). To test whether the rRNA mutations are sufficient to upregulate topAI translation, we expressed wild-type or mutant (23S rRNA ΔT1917) rRNA operons in trans in an otherwise wild-type strain, and measured the expression of the topAI-lux translational reporter fusion. For these and all additional assays of topAI-lux reporter fusions, we used a strain lacking topAI and yjhQ to rule out the possibility of autoregulation. Supplying the mutant 23S rRNA in trans resulted in increased topAI expression, whereas wild-type 23S rRNA did not (Figure 2B). Thus, the 23S rRNA mutation is dominant over the seven chromosomal rRNA operons, suggesting that the effect of the mutations in 23S rRNA genes on topAI expression is not simply due to a reduction in the levels of active ribosomes.

Expression of toxin–antitoxin genes is often induced by environmental stresses (Page and Peti, 2016). We next aimed to identify an environmental condition(s) that topAI responds to. Given the effect of mutating 23S rRNA on topAI expression, we speculated that perturbing translation with ribosome-targeting antibiotics might induce expression of topAI. Moreover, increases in topAI mRNA levels upon treatment with erythromycin or clindamycin have been reported previously (Dzyubak and Yap, 2016). We measured expression of the topAI translational reporter fusion in cells grown with each of 12 ribosome-targeting antibiotics at sub-inhibitory concentrations. Tetracycline, spectinomycin, retapamulin, tylosin, and erythromycin caused increases in expression from 250-fold to 4000-fold of

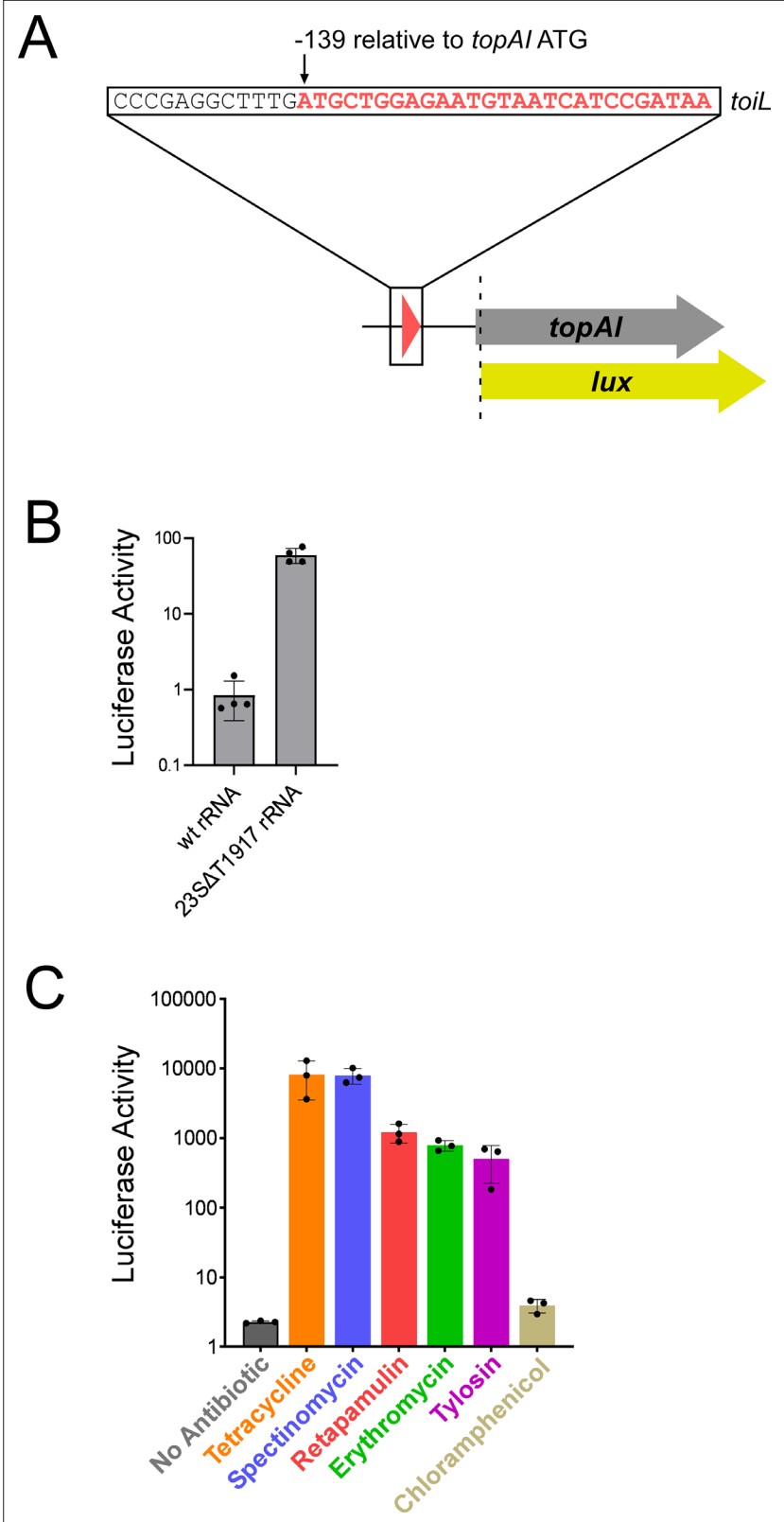

**Figure 2.** *topAI* expression is induced in response to translation stress. (**A**) Schematic representation of *topAI-lux* translational reporter fusions, indicating the position of the *toiL* start codon with respect to the *topAI* start codon. (**B**) Luciferase activity of the wild-type *topAI-lux* translational reporter fusion (pGB202) in Δ*topAI-yjhQ* cells (strain GB001) expressing either wild-type 23S rRNA (plasmid pGB322) or ΔT1917 23S rRNA (plasmid pGB318) in *trans*.

*Figure 2 continued on next page*

*Figure 2 continued*

Activity was measured 4 hours post-induction with propionate. (**C**) Luciferase activity of wild-type and mutant *topAI-lux* translational reporter fusions in Δ*topAI-yjhQ* cells (strain GB001) grown in the presence of ribosome-targeting antibiotics. Cells were grown in LB medium to an $OD_{600}$ of ~1.0. Indicated samples were treated with tetracycline (0.5 µg/ml), spectinomycin (90 µg/ml), retapamulin (4 µg/ml), erythromycin (100 µg/ml), tylosin (400 µg/ml), or chloramphenicol (1 µg/ml). Luminescence was measured 90 min after antibiotic treatment. Error bars represent ±1 standard deviation from the mean (n=3).

The online version of this article includes the following figure supplement(s) for figure 2:

**Figure supplement 1.** *topAI* is translationally repressed.

the *topAI-lux* translational fusion (***Figure 2C***; ***Supplementary file 1B***). We observed no change in expression in response to treatment with chloramphenicol (***Figure 2C***), kasugamycin, gentamicin, amikacin, streptomycin, apramycin, or hygromycin (***Supplementary file 1B***), although we cannot rule out that these antibiotics might induce *topAI* expression at concentrations other than those tested here. Indeed, a previous study showed that *topAI* RNA levels are strongly increased upon treatment with gentamicin (***Kohanski et al., 2008***). Another study showed that *topAI* levels are strongly induced upon treatment with kanamycin (***Shaw et al., 2003***), an antibiotic not tested in our work. In summary, *topAI* expression is modulated by a wide variety of mechanistically distinct ribosome-targeting antibiotics, reinforcing the idea that expression of this toxin–antitoxin system responds to the translation status of the cell.

To determine whether the relief of translation repression by ribosome-targeting antibiotics also prevents premature Rho-dependent transcription termination, we used ChIP-qPCR to measure the association of RNAP (β subunit) across the *topAI-yjhQP* operon in wild-type or *rho* mutant cells grown in the presence or absence of a sub-inhibitory concentration of tetracycline. As a control, we measured the association of RNAP within *rho*, where premature Rho-dependent transcription termination has been previously described (***Bastet et al., 2017***; ***Matsumoto et al., 1986***). For analysis of RNAP association within *rho*, we normalized RNAP occupancy values to those in the *rhoL* region upstream; for *topAI-yjhQP*, we normalized RNAP occupancy values within the transcribed regions to those in the promoter region (***Figure 3A***). Mutation of *rho* led to a significant 12.0-fold increase in RNAP occupancy within *rho* (*t*-test p=0.001), consistent with Rho-dependent transcription termination early in the *rho* transcript (***Figure 3B***). Tetracycline treatment in wild-type cells did not lead to a significant increase in RNAP occupancy within *rho* (*t*-test p=0.1). The combination of mutating *rho* and treating with tetracycline led to a significant 5.9-fold increase in RNAP occupancy within *rho* (*t*-test p=0.0009).

Mutation of *rho* led to significant 3.6-, 18.9-, and 17.8-fold increases in RNAP occupancy at three positions across the *topAI-yjhQP* operon (***Figure 3C***; *t*-test p=0.009, 0.004, and 0.001, respectively). These data are consistent with Rho-dependent transcription termination within *topAI*. The larger increases in RNAP occupancy at later positions within the transcribed region are likely due to a combination of the limited spatial resolution of ChIP-qPCR, and the potential for multiple termination sites within the transcribed region. Tetracycline treatment in wild-type cells led to significant 1.7-, 7.3-, and 6.7-fold increases in RNAP occupancy at the three positions across the *topAI-yjhQP* operon (*t*-test p=0.03, 0.0009, and 0.001, respectively), consistent with tetracycline treatment inhibiting Rho-dependent transcription termination within *topAI*. The combination of mutating *rho* and treating with tetracycline led to 2.0-, 5.7-, and 9.9-fold increases in RNAP occupancy at the three positions across the *topAI-yjhQP* operon (*t*-test p=0.0004, 0.006, 0.005, respectively).

To determine whether the relief of translation repression by ribosome-targeting antibiotics affects the *topAI-yjhQP* RNA level, we used qRT-PCR to measure RNA levels at positions across the *topAI-yjhQP* operon in wild-type or *rho* mutant cells grown in the presence or absence of a sub-inhibitory concentration of tetracycline. Mutation of *rho* led to significant 7.5-, 5-, and 8.4-fold increases in RNA levels at three positions across the *topAI-yjhQP* operon (***Figure 3D***; *t*-test p=0.002, 0.002, and $3.5e^{-7}$, respectively). These data are consistent with Rho-dependent transcription termination within *topAI*. Given that the most upstream region tested is upstream of the likely Rho termination site, these data suggest that premature Rho-dependent transcription termination destabilizes the *topAI-yjhQP* mRNA. Tetracycline treatment in wild-type cells led to significant 4.9-, 2.0-, and 8.0-fold increases in RNA levels at the three positions across the *topAI-yjhQP* operon (*t*-test p=0.008, 0.0007, and $8.2e^{-6}$, respectively), consistent with tetracycline treatment inhibiting Rho-dependent transcription

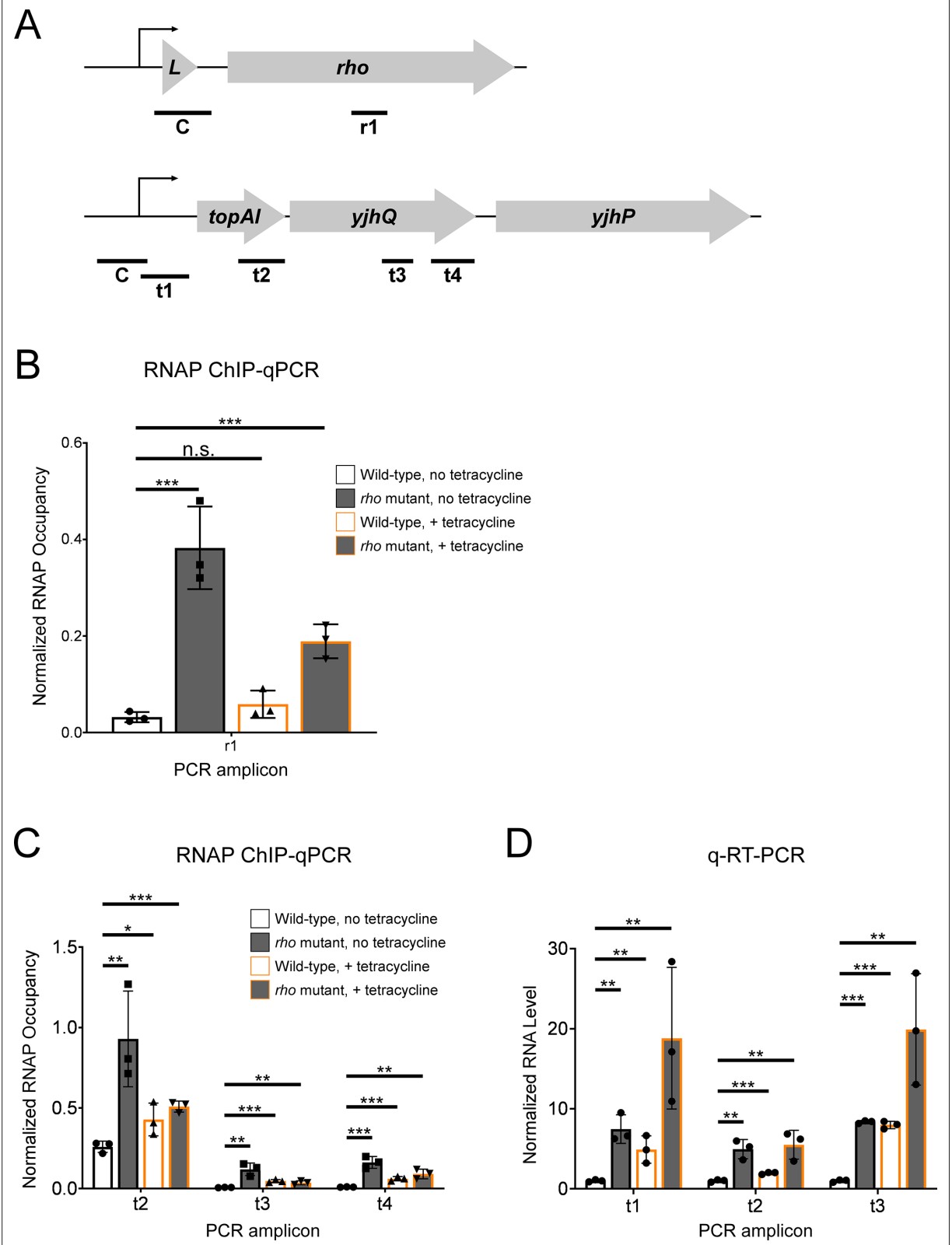

**Figure 3.** Tetracycline treatment reduces premature Rho-dependent transcription termination within *topAI-yjhQ* and increases abundance of the associated mRNA. (**A**) Schematic of the *rhoL-rho* and *topAI-yjhQP* operons showing the positions of PCR amplicons used for ChIP-qPCR and/or qRT-PCR. (**B**) Occupancy of RNAP measured by ChIP-qPCR in the *rho* gene. Occupancy values are normalized to those in *rhoL* (control PCR amplicon 'C' in panel A). Data are shown for wild-type (strain MG1655) or *rho* mutant (R66S amino acid substitution in Rho; strain CRB016) cells grown in the presence

*Figure 3 continued on next page*

*Figure 3 continued*

or absence of tetracycline, as indicated. The *x*-axis label corresponds to the PCR amplicon shown in panel A. (**C**) Occupancy of RNAP measured by ChIP-qPCR at positions across the *topAI-yjhQ* transcript. Occupancy values are normalized to those in the *topAI* upstream region (control PCR amplicon 'C' in panel A). (**D**) RNA levels measured by qRT-PCR at positions across the *topAI-yjhQ* transcript, normalized to the *mreB* gene. Error bars represent ±1 standard deviation from the mean (n=3). Statistical significance is indicated as follows: n.s., not significant (p>0.05), *p<0.05, **p<0.01, ***p<0.001.

termination within *topAI*. The combination of mutating *rho* and treating with tetracycline led to 18.8-, 5.5-, and 19.9-fold increases in RNA levels at the three positions across the *topAI-yjhQP* operon (*t*-test p=0.01, 0.006, and 0.005, respectively). Together, our data support a model in which tetracycline treatment induces translation of *topAI*, which in turn prevents premature Rho-dependent transcription termination within *topAI*.

## Expression of a uORF, *toiL*, is required for antibiotic-mediated *topAI* regulation

Long 5′ URs often contain regulatory elements such as uORFs that contribute to regulation of the downstream genes (**Kriner et al., 2016**). Given that *topAI* regulation responds to translation perturbation, we searched for a potential uORF that could act as a regulatory module in the *topAI* UR. Through manual inspection of available ribosome profiling data, we identified an eight-codon putative uORF that starts 139 nt upstream of *topAI* (**Figure 4A**; **Meydan et al., 2019**; **Wang et al., 2015**; **Weaver et al., 2019**). A small protein (*Mia*-127) with near-identical amino acid sequence and relative genome position was described in *Salmonella enterica* (**Figure 4—figure supplement 1**; **Baek et al., 2017**). To experimentally determine the frame and position of this uORF, which we renamed *toiL* (*topoisomerase inhibitor leader*) in *E. coli* K-12, we measured expression of a *toiL-lacZ* translational fusion. Replacing native codons with stop codons in the predicted CDS prevented expression of this fusion, unlike the equivalent substitution upstream of the ORF (**Figure 4**), supporting our ORF prediction. The first seven amino acids of the ToiL protein are conserved across bacteria that encode the *topAI-yjhQP* operon (**Figure 4—figure supplement 1**).

To determine if *toiL* is a regulatory uORF, we tested whether expression of the *topAI-lux* translational reporter fusion is induced by ribosome-targeting antibiotics if *toiL* translation is impaired. We first mutated the start codon of *toiL* from 'ATG' to 'gTa' (**Figure 5A**; the specific substitutions were made to preserve the predicted secondary structure of the 5′ UR; **Figure 5—figure supplement 1A**). The start codon mutation (**Figure 5A**, **Figure 5—figure supplement 1B**) did not impact expression of a *topAI-lux* transcriptional reporter fusion that used the entire UR for *topAI* (**Figure 5—figure supplement 1C**), indicating that the mutation does not cause premature Rho termination. We measured expression of the wild-type and mutant *topAI-lux* translational fusions in the presence or absence of sub-inhibitory concentrations of ribosome-targeting antibiotics. The *toiL* start codon mutation caused a 12- to 110-fold decrease in *topAI* induction by antibiotics (**Figure 5B**). Nonetheless, antibiotics were still able to modestly induce *topAI* expression. The *toiL* start codon is flanked by in-frame TTG and CTG trinucleotides that could potentially function as secondary start codons, perhaps explaining why mutation of the ATG start codon did not completely abolish the effect of antibiotic treatment. To test this possibility, we disrupted the predicted *toiL* Shine–Dalgarno (S-D) sequence (**Figure 5A**) and measured *topAI-lux* induction by ribosome-targeting antibiotics. Disruption of the *toiL* S-D site abolished induction of *topAI* expression by all antibiotics tested (**Figure 5B**), supporting the idea that active translation of *toiL* is absolutely required for *topAI* regulation by ribosome-targeting antibiotics.

## Tetracycline causes de-repression of *topAI* translation by disrupting base-pairing around the *topAI* ribosome-binding site

Long URs that encode uORFs often utilize alternating RNA structures to modulate expression of the downstream gene (**Kriner et al., 2016**). For *topAI*, the class of regulatory element (i.e., a uORF) and the type of inducer (i.e., ribosome-targeting antibiotics) closely resembles the regulation of rRNA methyltransferase genes *ermB/ermC* (**Subramanian et al., 2012**), where stalling of the ribosome within a uORF in the presence of certain macrolide antibiotics alters the downstream RNA structure, facilitating *ermB/ermC* translation. Computational prediction of the *topAI* UR RNA secondary structure suggests that sequences within *toiL* base-pair with the *topAI* ribosome-binding site (**Figure 6A**). Consistent with this prediction, the 5′ UR of *toiL* has been reported to form an extended region

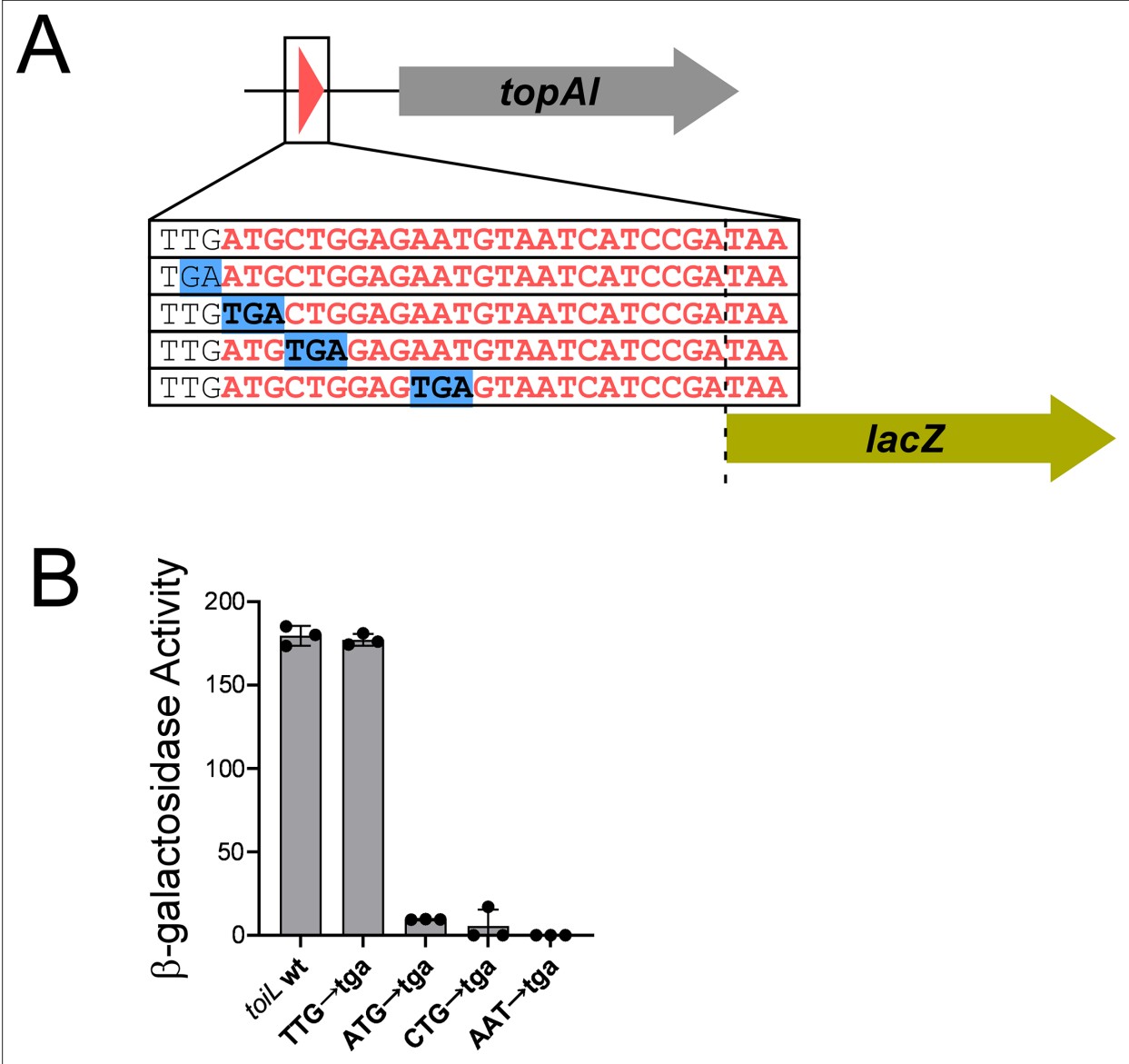

**Figure 4.** A uORF, *toiL*, is located within the *topAI* 5′ upstream region. (**A**) Schematic representation of *toiL-lacZ* translational reporter fusions, indicating mutations upstream and within the *toiL* uORF. (**B**) β-galactosidase activity of the wild-type *toiL-lacZ* translational reporter fusion (wt; plasmid pGB164), and fusions with mutations immediately upstream of *toiL* (TTG→tga; plasmid pGB201), at the *toiL* start codon (ATG→tga; plasmid pGB200), second codon (CTG→tga; plasmid pGB197), or fourth codon (AAT→tga; plasmid pGB196), in cells with wild-type *topAI-yjhQP* (strain AMD054). Error bars represent ±1 standard deviation from the mean (n=3).

The online version of this article includes the following figure supplement(s) for figure 4:

**Figure supplement 1.** ToiL conservation across *Enterobacteriaceae* species.

of dsRNA (*Lybecker et al., 2014*). Base-pairing between *toiL* and the *topAI* ribosome-binding site is further supported by a structural prediction that incorporates sequence conservation information from orthologous regions in related species (*Figure 6—figure supplement 1*).

We used in-cell SHAPE-seq (*Watters et al., 2016a*) to experimentally assess the *topAI* UR RNA secondary structure and investigate changes in secondary structure that occur upon tetracycline treatment. The SHAPE reagent, 1M7, penetrates live cells and modifies the backbone of accessible RNA nucleotides. In the subsequent RNA library preparation steps, modified nucleotides block reverse transcription (RT), creating RT-stop points that are detected bioinformatically after the 1M7-treated and untreated libraries are sequenced. Lower SHAPE reactivity is an indication of increased base-pairing interactions or nucleotide occlusion by other cellular factors such as ribosome binding. The

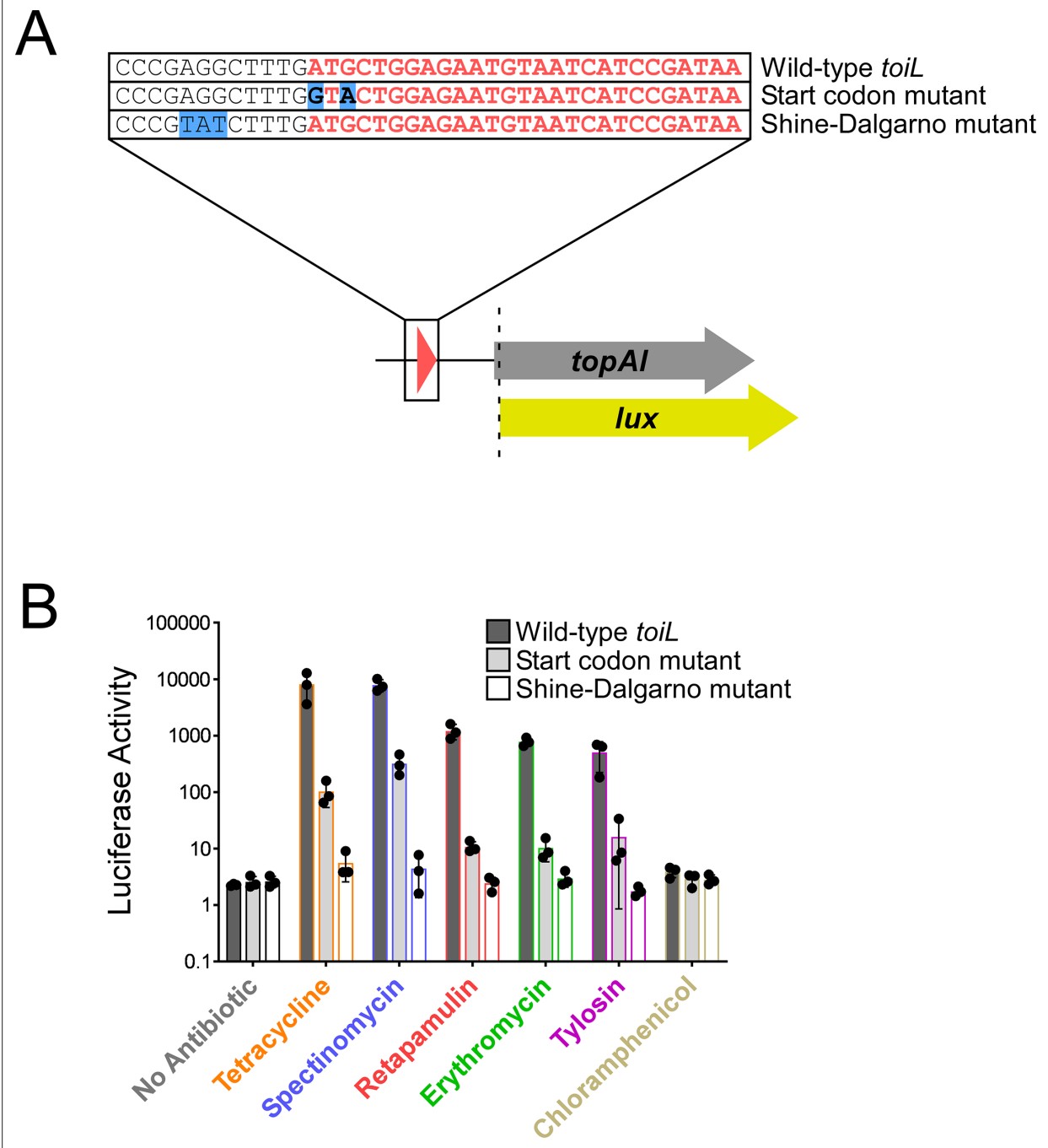

**Figure 5.** *toiL* translation is required for induction of *topAI* expression by ribosome-targeting antibiotics. (**A**) Schematic representation of *topAI-lux* translational reporter fusions, indicating mutations in the start codon and Shine–Dalgarno sequence of the *toiL* uORF. (**B**) Luciferase activity of wild-type and mutant *topAI-lux* translational reporter fusions in Δ*topAI-yjhQ* cells (strain GB001) grown in the presence of ribosome-targeting antibiotics. Cells were grown as indicated in *Figure 2C*. Dark gray bars show data for the wild-type *topAI-lux* translational reporter fusion (plasmid pGB202); light gray bars show data for the *topAI-lux* translational reporter fusion with a mutated *toiL* start codon (plasmid pGB313); white bars show data for the *topAI-lux* translational reporter fusion with a mutated *toiL* Shine–Dalgarno sequence (pGB366). Note that data for the wild-type *topAI-lux* translational reporter fusion are identical to those in *Figure 2C* and are included here for reference. Error bars represent ±1 standard deviation from the mean (n=3).

The online version of this article includes the following figure supplement(s) for figure 5:

**Figure supplement 1.** Mutation of the *toiL* start codon does not alter predicted RNA secondary structure and does not lead to Rho-dependent transcription termination in the *topAI* 5′ upstream region.

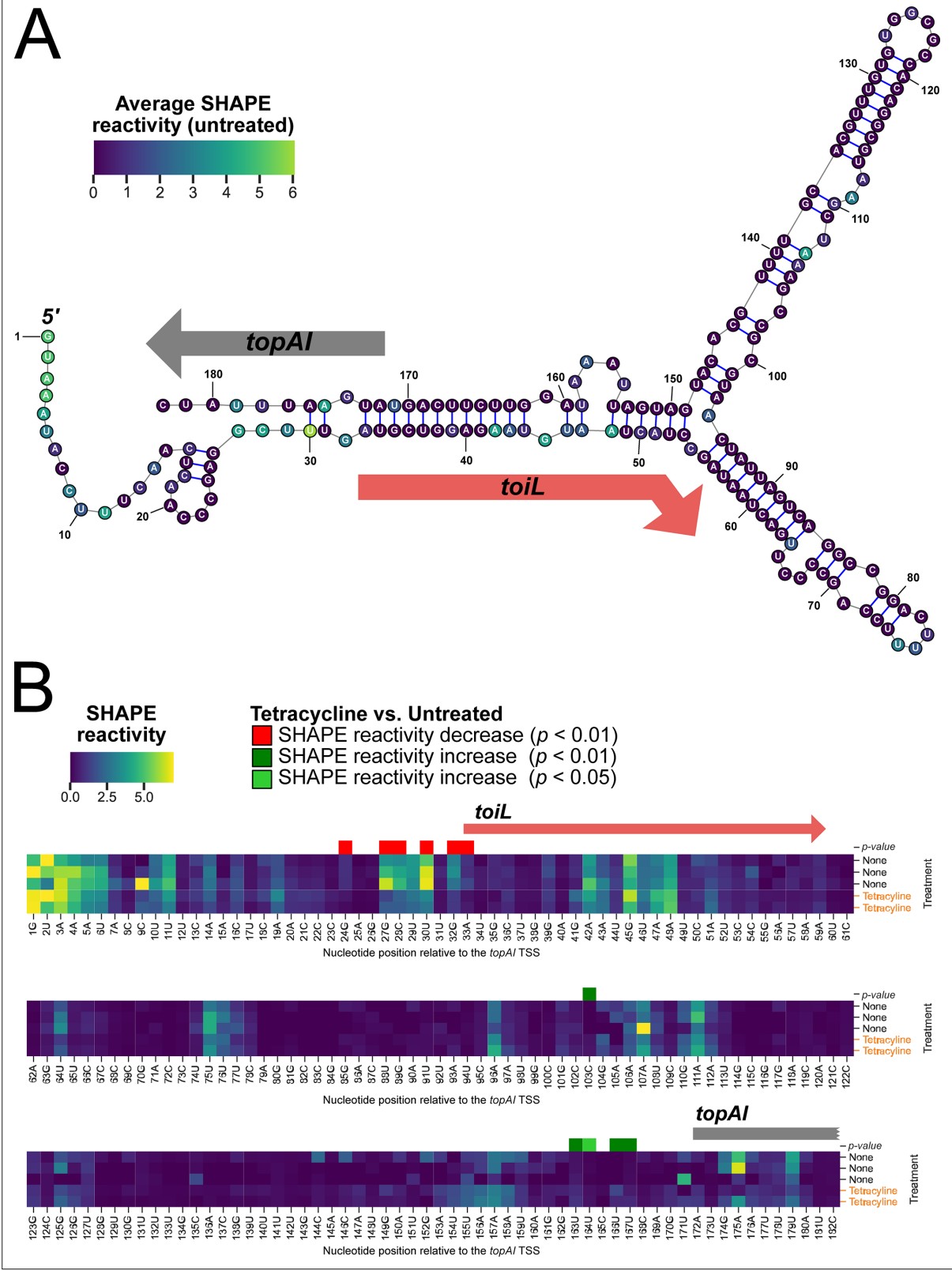

**Figure 6.** Structural changes in the *topAI* 5' upstream region induced by tetracycline treatment. (**A**) Predicted RNA secondary structure of the *topAI* 5' UR, indicating in-cell SHAPE reactivities of each base from an untreated sample. Reactivity scores are indicated on a color gradient. The position of the *toiL* uORF is indicated by a red arrow, and the position of the start of *topAI* is indicated by a gray arrow. Numbers indicate position relative to the mRNA 5' end. (**B**) In-cell SHAPE reactivities of each base from three replicates of untreated samples and two replicates of tetracycline-treated samples. SHAPE

*Figure 6 continued on next page*

*Figure 6 continued*

reactivity values are indicated on the same color gradient as panel A. Nucleotide positions with statistically significant differences (p<0.01 or 0.05, as indicated) in SHAPE reactivity between untreated and tetracycline-treated samples are indicated by red (lower reactivity in tetracycline-treated samples) or green (higher reactivity in tetracycline-treated samples) squares.

The online version of this article includes the following figure supplement(s) for figure 6:

**Figure supplement 1.** Predicted RNA secondary structure for the *topAI* upstream region based on sequence conservation.

SHAPE-seq data for untreated cells were in good agreement with the computationally predicted structure of the *topAI* UR (*Figure 6A*; *Supplementary file 1C*): the beginning of the 5' mRNA end is highly reactive, as are predicted loops. We speculate that moderate reactivity in the *toiL* coding region is the result of translating ribosomes that temporarily disrupt the predicted long-range base-pairing interactions. Tetracycline treatment significantly altered reactivity (p<0.05; see 'Materials and methods' for details) in two distinct regions in the 5' region in comparison to the untreated control: sequence from the *toiL* S-D to the *toiL* start codon became less reactive, whereas the *topAI* S-D became more reactive (*Figure 6B*; *Supplementary file 1C*). We propose that, in the presence of tetracycline, initiating ribosomes stall at the start of *toiL*, which leads to a decrease in base-pairing between *toiL* and the *topAI* S-D. No significant changes in the *toiL* ORF region were observed as a result of tetracycline treatment (p>0.05), likely due to these nucleotides switching from occlusion by base-pairing to occlusion by a stalled ribosome. The increase in reactivity around the *topAI* S-D following tetracycline treatment is statistically significant, but modest, suggesting that loss of base-pairing between *toiL* and this region is transient, and/or that a ribosome initiating translation of *topAI* partially prevents access to the SHAPE reagent.

## Tetracycline stalls ribosomes at the start codon of *toiL* and at start codons across the transcriptome

To directly measure ribosome positioning within *toiL* in vivo, we used ribosome profiling to map ribosome occupancy across the *E. coli* transcriptome in untreated cells, and cells treated with tetracycline. A comparison of ribosome occupancy across all genes showed that *topAI* and its operonic genes *yjhQ* and *yjhP* are among the most strongly induced by tetracycline (*Figure 7A*; *Supplementary file 1D*; note that induction could be at the level of RNA abundance, translation, or both). For cells treated with tetracycline, ribosome occupancy throughout *toiL* was substantially higher than in untreated cells. When mapping the 3' ends of ribosome-protected RNA fragments, we observed a strong peak at position +18 of *toiL*, a location consistent with the downstream edge of a ribosome stalled at the *toiL* start codon (*Figure 7B*). To determine whether tetracycline induces ribosome stalling in other ORFs, we compared ribosome occupancy in the region around the start codons of all annotated ORFs. Tetracycline induced stalling at positions +16 to +18 relative to start codons (*Figure 7C*), consistent with ribosomes stalled at start codons, as has been described for tetracycline in an earlier study (*Nakahigashi et al., 2016*). We conclude that tetracycline induces expression of *topAI* by stalling ribosomes at the start codon of *toiL*. We reasoned that retapamulin, a ribosome-targeting antibiotic that traps initiating ribosomes on start codons (*Meydan et al., 2019*; *Yan et al., 2006*), would induce *topAI* expression by the same mechanism. Consistent with this prediction, retapamulin strongly induced (~600-fold) expression of the *topAI-lux* translational reporter fusion (*Figure 2C*), and induction by retapamulin depended upon active translation of *toiL* (*Figure 5B*).

## Mapping ribosome stalling sites induced by different antibiotics

At least five different ribosome-targeting antibiotics induce expression of *topAI-yjhQP* (*Figure 2C*). We speculated that these antibiotics, like tetracycline, induce *topAI-yjhQP* expression by stalling ribosomes within *toiL*, preventing base-pairing between the *toiL* ORF and the *topAI* S-D. Moreover, because these antibiotics have different targets within the ribosome and a range of mechanisms of action, we speculated that stalling might occur at different positions in *toiL* depending on the antibiotic used. To further assess potential ribosome stalling at *toiL* in vivo, we used a modified version of a previously described stalling reporter construct (*Bailey et al., 2008*). Macrolide-mediated induction of this reporter relies on ribosome stalling at codon Ile9 of the *ermCL* uORF. We designed a hybrid construct where progressively longer segments of the *toiL* ORF, extending toward the 3' end, are fused

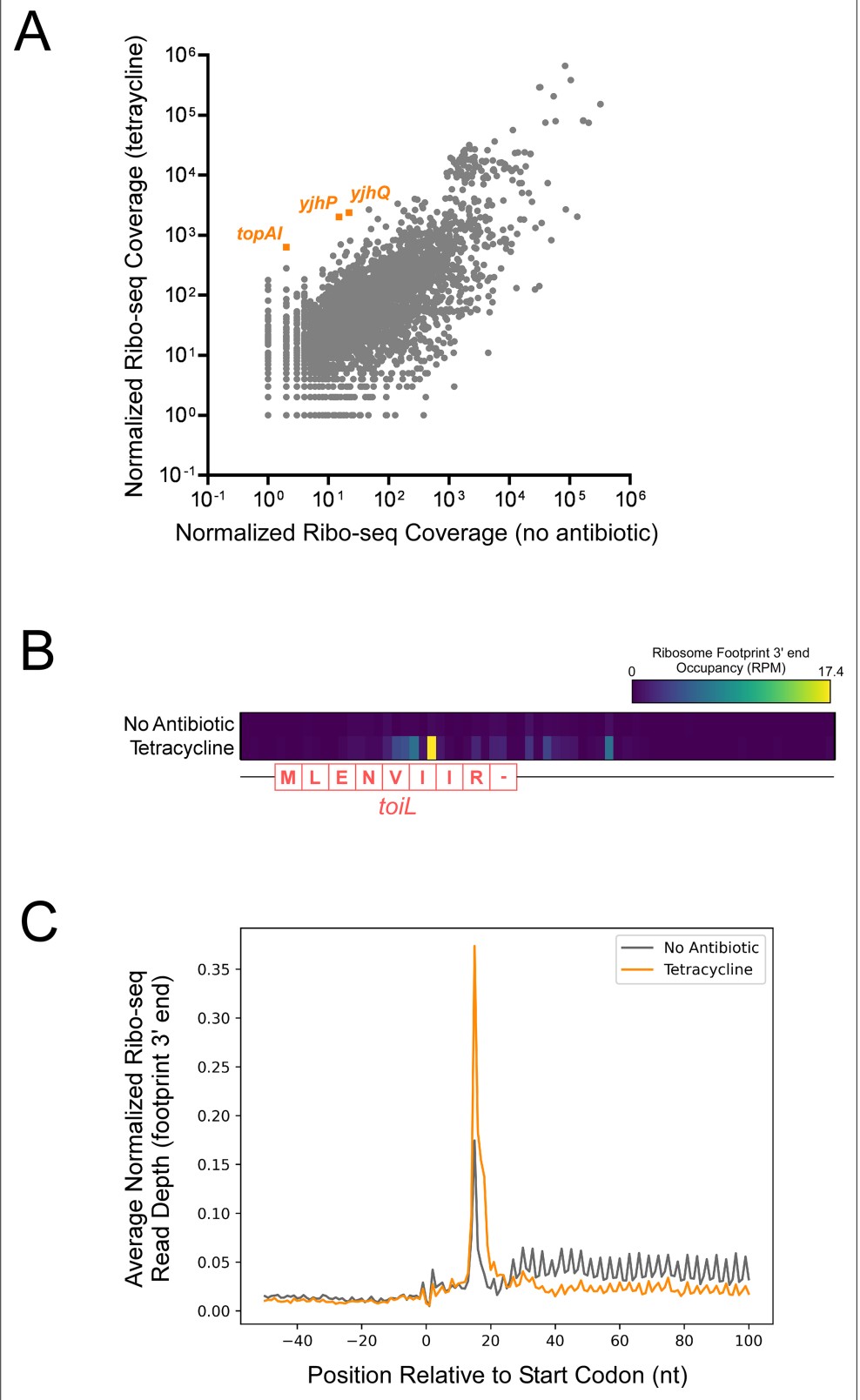

**Figure 7.** Tetracycline stalls ribosomes on start codons. (**A**) Normalized Ribo-seq coverage for all annotated ORFs for cells (strain MG1655) grown ± tetracycline. (**B**) Heatmap showing normalized Ribo-seq coverage in the region around the *toiL* uORF in untreated cells (no antibiotic) or tetracycline-treated cells. The color indicates the sequence-read coverage (reads per million; RPM) for RNA fragment 3′ ends that are presumed to represent the

*Figure 7 continued on next page*

*Figure 7 continued*

downstream edge of ribosome footprints. The *toiL* uORF position and encoded amino acid sequence is indicated. (**C**) Average of normalized Ribo-seq coverage for the regions around start codons for all annotated ORFs, for untreated cells (gray line; no antibiotic) or tetracycline-treated cells (orange line). Sequence-read coverage was calculated for RNA fragment 3' ends that are presumed to represent the downstream edge of ribosome footprints.

to the *ermCL* 10th codon and the remaining downstream sequence, followed by a luciferase reporter gene (*Figure 8A and B*; *Figure 8—figure supplement 1*). These constructs include the RNA structural features from the *ermCL-ermC* intergenic region that are altered in response to ribosome stalling upstream. Thus, ribosome stalling within *toiL* sequences, close to the junction with *ermCL*, is expected to induce expression of the luciferase reporter gene (*Figure 8A*, *Figure 8—figure supplement 1*). We measured expression of the stalling reporter constructs in a strain lacking *topAI* and *yjhQ*. As a control, we replaced *toiL* with the native *ermCL* sequence to show that this reporter is responsive to erythromycin treatment, but not other antibiotics (*Figure 8—figure supplement 2*). Tetracycline and retapamulin induced expression most strongly with the fusion to the start codon of *toiL* (*Figure 8C*), consistent with ribosome stalling at the start codon. Tylosin induced expression of only the 5-codon fusion reporter construct, suggesting tylosin-induced stalling at Val5. The effect of tylosin on expression of this reporter was abolished by the Val5→Leu mutation (*Figure 8C*). Interestingly, tetracycline, retapamulin, and erythromycin also showed some induction with the 5-codon *toiL* fusion, with the effect reduced by the Val5→Leu mutation (*Figure 8C*); additional stalling at this codon might have an additive effect on *topAI* induction. Erythromycin induced expression most strongly with the construct that has the complete *toiL* ORF, including the stop codon (*Figure 8C*), suggesting that ribosomes stall during translation termination. Spectinomycin induced expression of several constructs, but the level of induction was lower than for other antibiotics (*Figure 8C*), despite being able to induce *topAI* to a similar extent to tetracycline (*Figure 2C*). Chloramphenicol did not induce expression of any of the constructs, consistent with its inability to induce expression of *topAI* (*Figure 2C*). Overall, these data suggest that a variety of ribosome-targeting antibiotics cause ribosomes to stall within *toiL*, with different antibiotics stalling ribosomes at different positions. Moreover, the data suggest that ribosome stalling at a variety of positions within *toiL* is sufficient to induce *topAI-yjhQP* expression.

## Discussion

### Model for *topAI* regulation

We propose the following model for *topAI* regulation (*Figure 9*): In the absence of translation stress, a hairpin that encompasses the *toiL* uORF and the *topAI* ribosome-binding side prevents initiation of *topAI* translation, which in turn promotes premature Rho-dependent transcription termination within the *topAI* ORF. Under these repressive conditions, *toiL* is presumably being efficiently translated, but ribosomes translating *toiL* only transiently disrupt the hairpin. Upon induction of certain types of translation stress, notably ribosome inhibition by select antibiotics, ribosomes stall within *toiL*, disrupting the repressive hairpin and leading to translation of *topAI*, which in turn prevents Rho-dependent transcription termination within *topAI*. *topAI-yjhQP* RNA levels in the *rho* R66S mutant increase approximately twofold upon tetracycline treatment (*Figure 3D*), suggesting an additional, albeit modest level of regulation at the level of mRNA stability. This could be due to a stabilizing effect of translation (*Deana and Belasco, 2005*) or direct regulation of RNase accessibility in the 5' UR.

### *toiL* is a multipurpose sensory uORF

Sensory uORFs are versatile regulators that have evolved to rapidly alter physiology in response to a wide variety of environmental stresses that impact translation. However, most regulatory uORFs characterized to date have been proposed to respond to a single translational stress. By contrast, *toiL* senses a wide variety of ribosome-targeting antibiotics with unrelated mechanisms of action. The ability of *toiL* to transduce a wide range of translational stress signals is likely due to the fact that base-pairing with the *topAI* ribosome-binding site occurs across the length of the *toiL* ORF; hence, ribosome stalling anywhere within *toiL* would likely prevent hairpin formation, relieving *topAI* repression. This likely explains why mutations in helix 69 and helix 80 of the 23S rRNA induce expression of *topAI* since residues in these helices have been reported to promote ribosome processivity (*Kipper*

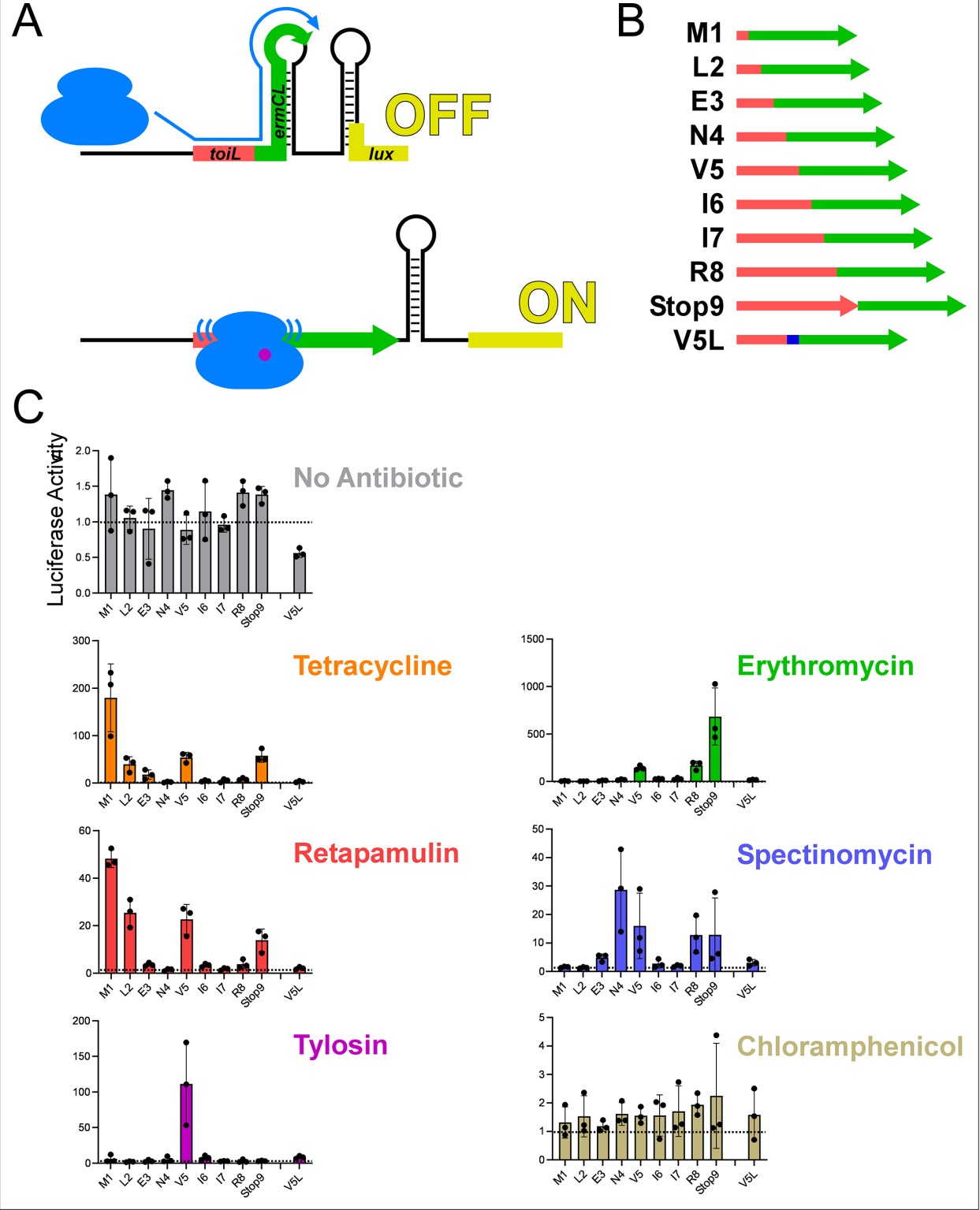

**Figure 8.** Ribosome stalling at the *topAI* leader in vivo. (**A**) Schematic showing the *toiL-ermCL-lux* stalling reporter in the translationally inactive conformation ('OFF') that is expected in the absence of ribosome stalling, and the translationally active conformation ('ON') that is expected when ribosomes stall within *toiL* sequence close to the junction of *toiL* with *ermCL*. (**B**) Schematic showing *toiL-ermCL* stalling reporters where *toiL* is progressively extended by one codon at the 3′ end and fused to part of *ermCL*. *toiL* sequence is indicated by red rectangles/arrow; *ermCL* sequence is indicated by green arrows. The last indicated stalling reporter extends to position 5 of *toiL* but has a Val5→Leu substitution (blue). (**C**) Luciferase activity of the *toiL-ermCL-lux* stalling reporters (plasmids pGB323, pGB346, pGB325, pGB326, pGB324, pGB328, pGB329, pGB307, pGB347, and pGB327)

*Figure 8 continued on next page*

*Figure 8 continued*

in Δ*topAI-yjhQ* cells (strain GB001). Cells were grown to an OD$_{600}$ of ~1.0 and treated with the indicated ribosome-targeting antibiotics at the same concentrations as in *Figure 2C*. Luminescence was measured three hours post-treatment. Horizontal dashed lines indicate luminescence activity of 1, to facilitate comparison between panels with different *y*-axis scales. Error bars represent ±1 standard deviation from the mean (n=3).

The online version of this article includes the following figure supplement(s) for figure 8:

**Figure supplement 1.** Predicted RNA structures for translationally inactive and active conformations of *toiL* stalling reporters.

**Figure supplement 2.** Expression of an *ermCL* stalling reporter is specifically induced by erythromycin.

*et al., 2009*; *Samaha et al., 1995*), and the impact of ribosomes slowing as they translate *toiL* could be similar to that of a ribosome stall. Many antibiotics have been shown to stall ribosomes in ORFs in a sequence-specific manner (*Orelle et al., 2013*; *Vázquez-Laslop and Mankin, 2018a*). We speculate that the *toiL* sequence has evolved to stall ribosomes in a sequence-specific manner in response to a variety of antibiotics.

There are a few other examples of uORFs where more than one class of antibiotic has been reported to induce ribosome stalling. *Clostridiodes difficile cplR* is regulated by a uORF that senses retapamulin and lincomycin (*Obana et al., 2023*). *Bacillus subtilis vmlR* is regulated by a uORF that senses retapamulin, lincomycin, and iboxamycin (*Takada et al., 2022*). *Staphylococcus haemolyticus vga(A)* is regulated by a uORF that senses lincosamides, streptogramin A, and pleuromutilins (*Vimberg et al., 2020*). However, in these cases, the antibiotics all likely function to stall ribosomes at start codons (*Takada et al., 2022*). Hence, even though the antibiotics have different targets within the ribosome, the position of ribosome stalling is the likely to be the same. By contrast, our data strongly suggest that while tetracycline and retapamulin stall ribosomes at the *toiL* start codon, tylosin, spectinomycin, and erythromycin stall ribosomes further downstream.

Another uORF that is known to sense multiple classes of ribosome-targeting antibiotics is found upstream of the actinobacterial gene *whiB7* (*Burian and Thompson, 2018*; *Lee et al., 2022*). Whether antibiotics induce ribosome stalling within the *whiB7*-associated uORF has not been tested. Intriguingly,

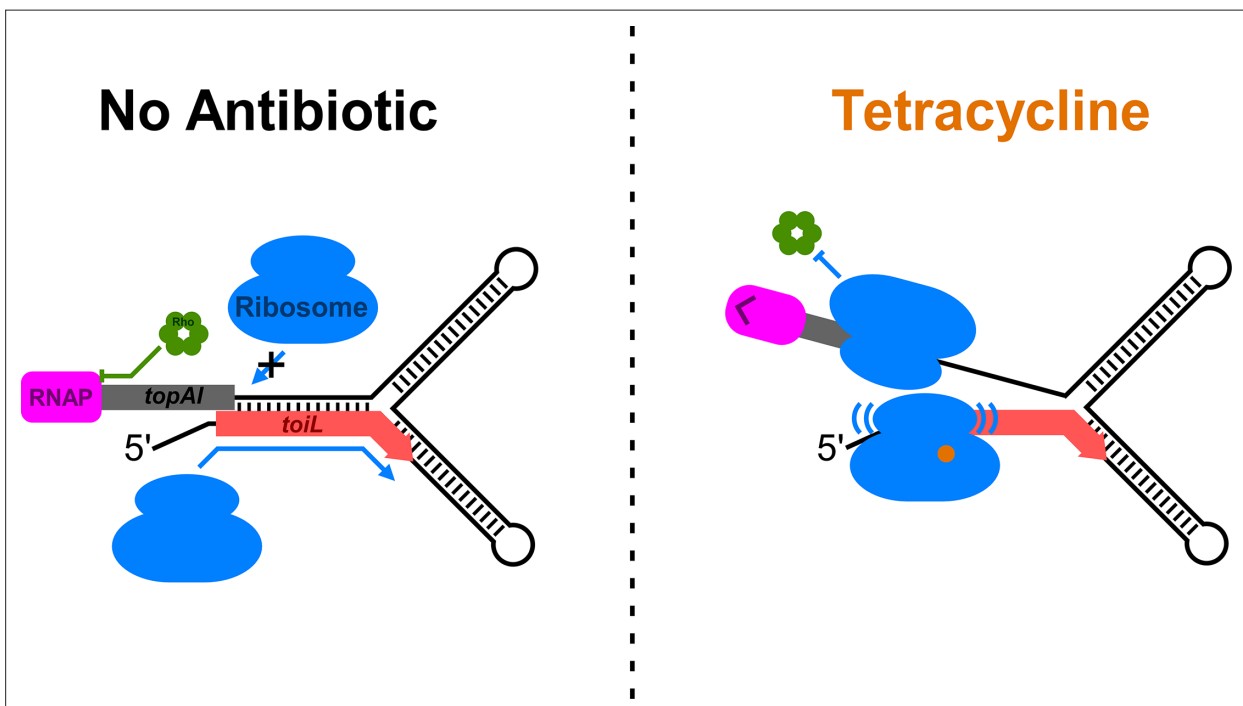

**Figure 9.** Model for regulation of *topAI* expression. Schematic showing a model for regulation of *topAI* expression. In the repressed state, a hairpin forms between the *toiL* sequence and the ribosome-binding site of *topAI*, repressing translation, which in turn promotes premature Rho-dependent transcription termination within the *topAI* gene. Ribosomes do not stall within *toiL*, so there is minimal perturbation of the repressive hairpin. By contrast, when cells are treated with certain ribosome-targeting antibiotics such as tetracycline, ribosomes stall within *toiL* in a sequence-specific manner, preventing formation of the repressive hairpin.

*whiB7* expression can also be induced by amino acid starvation (*Lee et al., 2022*), suggesting that non-antibiotic stresses that lead to ribosome stalling within the uORF can also contribute to regulation. An artificial uORF-regulated reporter construct has also been generated that responds to a variety of ribosome-targeting antibiotics, albeit to lesser extent than *topAI* or *whiB7*; the mechanism of uORF regulation in response to antibiotic treatment was presumed to involve ribosome stalling in the uORF, although this has not been tested (*Osterman et al., 2012*).

## Understanding how different antibiotics stall ribosomes within *toiL*

The mechanism by which tetracycline stalls ribosomes at start codons has not been determined. Tetracycline binds to the 30S ribosomal subunit in the decoding center and likely functions by preventing association of aminoacyl-tRNAs with the A-site (*Nguyen et al., 2014*). We speculate that tetracycline binds to initiating ribosomes when the A-site is free, preventing association of an aminoacyl-tRNA in the A-site, an essential step prior to formation of the first peptide bond in the nascent peptide. Intriguingly, regulation of tetracycline resistance can involve a uORF (*Chopra and Roberts, 2001*), including one case where the uORF is only four codons long (*Wang et al., 2005*), consistent with regulation by ribosome stalling at the start codon.

Macrolides such as erythromycin function by binding in the nascent peptide exit tunnel of the ribosome (*Vázquez-Laslop and Mankin, 2018b*). Ribosome stalling induced by erythromycin is well-established for uORFs upstream of macrolide-resistance genes (*Ramu et al., 2009*), where the nascent peptide in the exit tunnel senses the presence of erythromycin, leading to structural changes that prevent peptide bond formation. Our data are consistent with erythromycin inducing ribosome stalling at the end of *toiL* (*Figure 8C*), suggesting that the nascent peptide senses erythromycin, leading to structural changes that inhibit the action of release factors. Indeed, erythromycin was recently shown to stall ribosomes at the penultimate codon of the *Streptococcus msrDL* uORF by altering the 23S rRNA structure to prevent release factor binding (*Fostier et al., 2023*). When ribosomes stall on *msrDL* in the presence of erythromycin, the $N_{-3}$ and $N_{-2}$ amino acids with respect to the C-terminal residue are leucine and isoleucine respectively, and the stop codon is UAA. The $N_{-3}$ and $N_{-2}$ amino acids are the closest to erythromycin in the peptide exit tunnel (*Fostier et al., 2023*). Assuming erythromycin induces stalling of ribosomes on the penultimate codon of *toiL*, the $N_{-3}$ and $N_{-2}$ amino acids amino acids would be valine and isoleucine, respectively, and the stop codon is UAA, suggesting a similar interaction of the ToiL and MsrDL nascent peptides with erythromycin, and a similar mechanism of stalling.

Tylosin is also a macrolide, and stalling induced by tylosin has also been observed previously at an RYR sequence in *Bacillus subtilis* (*Yakhnin et al., 2019*). This suggests a similar mechanism of action to erythromycin, which can stall ribosomes at R/K-X-R/K motifs (*Davis et al., 2014*; *Ramu et al., 2009*). By contrast, our data suggest that tylosin induces ribosome stalling at Val5 of *toiL* (*Figure 8C*). Despite the apparent preference for R/K-X-R/K motifs for tylosin-induced ribosome stalling, tylosin has been observed to stall ribosomes at other sequences (*Orelle et al., 2013*), suggesting that the sequence requirements for tylosin-induced stalling are complex, as they are for erythromycin (*Ramu et al., 2009*).

Spectinomycin also induces expression of *topAI* in a *toiL*-dependent manner, although our data suggest that spectinomycin does not lead to ribosome stalling at a single codon position in *toiL* (*Figure 8C*). Little is known about the effects of spectinomycin on ribosome stalling, but it has been shown to stall ribosomes at specific sequences within ORFs (*Orelle et al., 2013*).

We did not observe any induction of *topAI* expression by chloramphenicol. Previous studies have shown that chloramphenicol stalls ribosomes in a sequence-specific manner, with stalling most pronounced immediately following an alanine, serine, or threonine codon (*Marks et al., 2016*; *Syroegin et al., 2022*). The lack of alanine, serine, and threonine codons in *toiL* may explain the apparent lack of stalling induced by chloramphenicol.

## The link between translation attenuation and premature Rho-dependent termination

In addition to directly regulating translation of *topAI*, *toiL* indirectly regulates *topAI* transcription by modulating Rho-dependent transcription termination within the *topAI* ORF. We speculate that Rho prematurely terminates transcription of other uORF-regulated genes where the uORF modulates

translation, similar to the function of some riboswitches (*Bastet et al., 2018*). Other studies of premature Rho termination events within coding regions suggested that termination occurs due to the unmasking of Ruts within the ORF that are otherwise occluded by translating ribosomes (*Bastet et al., 2017*; *Ben-Zvi et al., 2019*; *Bossi et al., 2012*; *de Smit et al., 2008*). By contrast, the Rut for *topAI* is located within the 5′ UR. This suggests that the mechanism by which translation of *topAI* prevents Rho termination is distinct to that of other characterized examples of prematurely Rho-terminated genes. Specifically, ribosomes translating *topAI* likely prevent Rho from catching the elongating RNAP, rather than preventing Rho loading onto the nascent RNA. If there are indeed two distinct mechanisms by which ribosomes can prevent Rho termination, the level of translation required for each mechanism may be different. An alternative possibility is that the *topAI* Rut is very long and extends into the *topAI* ORF, in which case ribosomes could prevent Rho termination by preventing Rho loading.

We identified elements of the *topAI* Rut that are located a short distance upstream of *toiL*. Given the length of RNA required to constitute a Rut, the *topAI* Rut must overlap *toiL*. Thus, the Rut is functional despite the presence of an overlapping ORF that is actively translated. We propose that (i) Rho can only load onto the *topAI* 5′ region after translation of *toiL* is terminated, or (ii) Rho can loop the RNA around the *toiL* region to access upstream and downstream elements of the Rut. It has previously been proposed that Rho can step over an RNA roadblock to access available Ruts (*Kriner and Groisman, 2017*; *Schwartz et al., 2007*), thereby helping Rho to overcome roadblocks in the 5′ region.

## Concluding remarks

uORFs permit rapid regulation in response to translational stress. For almost all previous uORFs shown to sense antibiotics, the downstream gene is functionally related to antibiotic resistance, often directly promoting resistance to the antibiotic(s) that induce expression. Although *topAI* has been characterized as a DNA gyrase inhibitor (*Yamaguchi and Inouye, 2015*), we speculate that *topAI* and/or *yjhQ/P*, which are transcribed from the same operon, have additional functions that are linked to the small molecule that *toiL* has evolved to sense. This small molecule could be a ribosome-targeting antibiotic, or a non-antibiotic molecule such as an amino acid or polyamine, which promote ribosome stalling in other uORFs (*Herrero Del Valle et al., 2020*; *van der Stel et al., 2021*).

There has been a recent explosion in the discovery of bacterial sORFs, many of which are located short distances upstream of canonical ORFs on the same strand (*Meydan et al., 2019*; *Smith et al., 2022*; *Stringer et al., 2021*; *Venturini et al., 2020*; *Weaver et al., 2019*). We speculate that there are large numbers of regulatory uORFs even in a well-characterized bacterium like *E. coli*. Indeed, analysis of ribosome profiling data for cells treated with tetracycline revealed >5-fold increases in ribosome-associated RNA levels relative to untreated cells for several genes that are known to be associated with a uORF (*Supplementary file 1D*), including *mgtA, corA,* and *speC*.

# Materials and methods

**Key resources table**

| Reagent type (species) or resource | Designation | Source or reference | Identifiers | Additional information |
|---|---|---|---|---|
| Strain, strain background (*Escherichia coli*) | MG1655 (WT) | This paper | N/A | See *Supplementary file 1E* for derivatives |
| Antibody | Mouse monoclonal Inti-*E. coli* RNA Polymerase β | BioLegend | 663003, clone: NT63; RRID:AB_2564516 | ChIP (1:800) |
| Recombinant DNA reagent | pAMD-BA-lacZ | *Stringer et al., 2014* | N/A | See *Supplementary file 1E* for derivatives |
| Recombinant DNA reagent | pPro24 | Addgene | RRID:Addgene_17805 | See *Supplementary file 1E* for derivatives |
| Recombinant DNA reagent | pET-PesaR-lux | Addgene | RRID:Addgene_47802 | See *Supplementary file 1E* for derivatives |
| Recombinant DNA reagent | pCS-PesaR-lux | Addgene | RRID:Addgene_47655 | See *Supplementary file 1E* for derivatives |

*Continued on next page*

*Continued*

| Reagent type (species) or resource | Designation | Source or reference | Identifiers | Additional information |
|---|---|---|---|---|
| Sequence-based reagent | Primers and geneblocks | IDT | N/A | See *Supplementary file 1E* for derivatives |
| Chemical compound, drug | Tetracycline | Sigma-Aldrich | T3383 | |
| Chemical compound, drug | Spectinomycin | Sigma-Aldrich | S4014 | |
| Chemical compound, drug | Erythromycin | Sigma-Aldrich | E5389 | |
| Chemical compound, drug | Retapamulin | Sigma-Aldrich | CDS023386 | |
| Chemical compound, drug | Tylosin | Sigma-Aldrich | T6134 | |
| Chemical compound, drug | Chloramphenicol | Sigma-Aldrich | C0378 | |
| Chemical compound, drug | Gentamicin | Sigma-Aldrich | G1272 | |
| Chemical compound, drug | Hygromycin B | Millipore | 400052 | |
| Chemical compound, drug | Sodium propionate | Sigma-Aldrich | P1880 | |
| Chemical compound, drug | 1M7 | MedChem Express | HY-D0913 | |
| Commercial assay or kit | Quick-RNA Miniprep Kit | Zymo Research | R1054 | |
| Commercial assay or kit | MultiScribe Reverse Transcriptase | Invitrogen | 4311235 | |
| Commercial assay or kit | SuperScript III Reverse Transcriptase | Invitrogen | 18080093 | |
| Commercial assay or kit | DNeasy Blood & Tissue Kit | QIAGEN | 69504 | |
| Commercial assay or kit | NEBuilder HiFi DNA Assembly Master Mix | NEB | E2621S | |
| Other | SphI-HF | NEB | R3182S | Restriction enzyme used for cloning plasmids |
| Other | BamHI-HF | NEB | R3136S | Restriction enzyme used for cloning plasmids |
| Other | HindIII-HF | NEB | R3104S | Restriction enzyme used for cloning plasmids |
| Other | XhoI-HF | NEB | R0146S | Restriction enzyme used for cloning plasmids |
| Other | XbaI-HF | NEB | R0145S | Restriction enzyme used for cloning plasmids |
| Other | EcoRI-HF | NEB | R3101S | Restriction enzyme used for cloning plasmids |
| Other | NheI-HF | NEB | R3131S | Restriction enzyme used for cloning plasmids |
| Other | SHAPE-seq reagents | *Watters et al., 2016a* | N/A | Used for experiments in *Figure 6* |
| Other | Ribo-seq reagents | *Stringer et al., 2021* | N/A | Used for experiment in *Figure 7* |

*Continued on next page*

*Continued*

| Reagent type (species) or resource | Designation | Source or reference | Identifiers | Additional information |
|---|---|---|---|---|
| Software, algorithm | Python | Python Language Reference | RRID:SCR_008394 | v3.7; https://www.python.org/ |
| Software, algorithm | seaborn | https://doi.org/10.21105/joss.03021 | RRID:SCR_018132 | v0.11.1; https://seaborn.pydata.org/index.html |
| Software, algorithm | R | The R Project | RRID:SCR_001905 | v4.1.2; https://www.r-project.org/ |
| Software, algorithm | DESeq2 | *Love et al., 2014* | RRID:SCR_015687 | v1.34.0; https://bioconductor.org/packages/release/bioc/html/DESeq2.html |
| Software, algorithm | CLC Genomics Workbench | QIAGEN | | v8.5.1; SNP calling |
| Software, algorithm | spats | *Watters et al., 2016a*; *Trapnell et al., 2017* | | v1.0.2; https://luckslab.github.io/spats/ |
| Software, algorithm | cutadapt | https://doi.org/10.14806/ej.17.1.200 | RRID:SCR_011841 | v0.12.8; https://cutadapt.readthedocs.io/en/stable/ |
| Software, algorithm | bowtie | https://doi.org/10.1186/gb-2009-10-3-r25 | RRID:SCR_005476 | v0.12.8; http://bowtie.cbcb.umd.edu/ |
| Software, algorithm | Rockhopper | *McClure et al., 2013* | | v2.0.3; https://cs.wellesley.edu/~btjaden/Rockhopper/index.html |
| Software, algorithm | ClustalO | *Madeira et al., 2022* | RRID:SCR_001591 | https://www.ebi.ac.uk/jdispatcher/msa/clustalo |
| Software, algorithm | Prism | GraphPad | RRID:SCR_002798 | v9.4.1 |
| Software, algorithm | PETfold | *Seemann et al., 2011* | | v2.2; https://rth.dk/resources/petfold/submit.php |
| Software, algorithm | Mfold | *Zuker, 2003* | RRID:SCR_008543 | v4.7; https://www.unafold.org/mfold/applications/rna-folding-form.php |
| Software, algorithm | RNAStructure | https://doi.org/10.1186/1471-2105-11-129 | RRID:SCR_017216 | v6.1; https://rna.urmc.rochester.edu/RNAstructure.html |

## Strains and plasmids

All strains and plasmids used in this study are listed in *Supplementary file 1E*. All oligonucleotides used in this study are listed in *Supplementary file 1F*. An *E. coli* MG1655 Δ*lacZ* Δ*topAI-yjhQ::thyA* strain (GB001) was constructed using the FRUIT recombineering method (*Stringer et al., 2012*). Briefly, the *thyA* gene was amplified using primers JW7676+JW7677 and electroporated into strain AMD189 (MG1655 Δ*lacZ* Δ*thyA*) (*Stringer et al., 2012*) to replace the *topAI* gene from 379 bp upstream of *topAI* to 74 bp into the *yjhQ* coding region.

All *topAI* and *toiL lacZ* and luciferase reporter plasmids (pJTW100, pGB164, pGB182, pGB196, pGB197, pGB200, pGB201, pGB202, pGB215, pGB217, pGB297, pGB305, pGB306, pGB313) were made to include sequence starting at –400 bp upstream of *topAI* and were presumed to include the native *topAI* promoter. Fusions to *lacZ* are derivatives of plasmid pAMD033 (*Dornenburg et al., 2010*). Fusions to luciferase are derivatives of pGB135, which includes the *luxCDABE* operon from *Photorhabdus luminescens*. Constructing pGB135 first required construction of plasmid pGB96. pGB96 was made by PCR-amplification of a region from plasmid pJTW064 (*Stringer et al., 2014*) using primers JW8044+JW8045, PCR-amplification of a region from pCS-PesaRlux (*Shong and Collins, 2013*) using primers JW8046+JW8047, combining the two PCR products using splicing by overlap extension (*Horton et al., 1990*), and cloning the resultant product into pET-PesaRlux cut with *XhoI* and *XbaI* (*Shong and Collins, 2013*). pGB135 was made by PCR-amplification of two regions from pGB96 using primers JW8044+JW8522 and JW8523+JW8047, combining the two PCR products using splicing by overlap extension (*Horton et al., 1990*), and cloning the resultant product into pCS-PesaRlux cut

with *Xho*I and *Xba*I (*Shong and Collins, 2013*). pGB135 has the P*esaR* fragment from pCS-PesaRlux replaced with a constitutive promoter and convenient restriction sites for transcriptional or translation fusions to the *luxC* gene; transcriptional reporter fusions included a Shine–Dalgarno sequence, whereas translational fusions used the sequence up to the initiation codon of the gene of interest followed by the second codon of the reporter gene.

Plasmid pJTW100 is a derivative of pAMD033 that includes a full-length *topAI* gene and the first 90 bp of *yjhQ* translationally fused to *lacZ*. pJTW100 was constructed by PCR-amplifying from an *E. coli* MG1655 colony using primers JW5638+JW5639, and cloning the resultant PCR product into pAMD033 (*Dornenburg et al., 2010*) cut with *Sph*I and *Hind*III.

Plasmids pGB215 and pGB217 are derivatives of pAMD033 with sequence up to position +10 or +42 of *topAI* (relative to the start codon) fused transcriptionally to *lacZ*. pGB215 and pGB217 were made by PCR-amplification from an *E. coli* MG1655 colony using primers JW5638+JW9453 or JW5638+JW9154, respectively, and cloning of the resultant PCR products into the *Sph*I and *Nhe*I sites of pAMD033 (*Dornenburg et al., 2010*). Plasmids pGB305 and pGB306 are equivalent to plasmids pGB215 and pGB217, except that they have a mutated *rut* in the 5′ UR (4C → 4A). They were constructed identically to pGB215 and pGB217 except that the PCRs used primers JW5638+JW9453/JW9154, and the template for the PCR was plasmid pGB299. Plasmid pGB299 is a derivative of pAMD033 (*Dornenburg et al., 2010*) with the complete *topAI* gene (not including the stop codon) fused translationally to *lacZ*, with a mutated *rut* in the 5′ UR (4C → 4A). pGB299 was constructed by PCR-amplifying from an *E. coli* MG1655 colony using primers JW5638+JW9803 and JW9804+JW8023, combining the two PCR products using splicing by overlap extension (*Horton et al., 1990*), and cloning the resultant product into pAMD033 cut with *Sph*I and *Hind*III.

Plasmid pGB214 is equivalent to pJTW100 except that it has a mutation in the *toiL* start codon (ATG→gTa). pGB214 was constructed by PCR-amplifying from an *E. coli* MG1655 colony using primers JW5638+JW9139 and JW9138+JW5639, combining the two PCR products using splicing by overlap extension (*Horton et al., 1990*), and cloning the resultant PCR product into pAMD033 (*Dornenburg et al., 2010*) cut with *Sph*I and *Hind*III. Plasmids pGB182 (wild-type) and pGB297 (*toiL* ATG→gTa) include the complete *topAI* 5′ UR transcriptionally fused to *lacZ*. pGB182 and pGB297 were made by PCR-amplifying from an *E. coli* MG1655 colony or pGB214, respectively, using primers JW5638+JW8809, and cloning the resultant PCR product into pAMD033 (*Dornenburg et al., 2010*) cut with *Sph*I and *Nhe*I.

Plasmid pGB164 is a derivative of pAMD033 with the full *toiL* gene (not including the stop codon) fused translationally to *lacZ*. Plasmids pGB196 (AAT→tga), pGB197 (CTG→tga), pGB200 (ATG→tga), and pGB201 (TTG→tga) are mutant derivatives of pGB164. pGB164 was made by PCR-amplifying from an *E. coli* MG1655 colony using primers JW5638+JW8741, and cloning the resultant PCR product into pAMD033 (*Dornenburg et al., 2010*) cut with *Sph*I and *Hind*III. pGB196, pGB197, pGB200, and pGB201 were cloned identically except that primer JW8741 was replaced with JW8999, JW8998, JW9013, and JW9014, respectively.

Plasmid pGB202 is a derivative of pGB135 with a translational fusion of the *topAI* start codon to luciferase. pGB313 is a derivative of pGB202 with a *toiL* start codon mutation. pGB202 and pGB313 were made by PCR-amplifying from an *E. coli* MG1655 colony or pGB214, respectively, using primers JW9288+JW9289, and cloning the resultant PCR product into pGB135 cut with *Bam*HI and *Eco*RI. Plasmid pGB366 is a derivative of pGB202 with a *toiL* Shine–Dalgarno sequence mutation. pGB366 was made by PCR-amplifying from an *E. coli* MG1655 colony using primers JW9288+JW10873 and JW10874+JW9289, combining the two PCR products using splicing by overlap extension (*Horton et al., 1990*), and cloning the resultant product into pGB135 cut with *Bam*HI and *Eco*RI.

Plasmids pGB323, pGB324, pGB325, pGB326, pGB327, pGB328, pGB329, pGB346, and pGB347 include sequences that extend to different positions within *toiL* fused translationally to the *ermCL* gene from the 10th codon, which is fused translationally to luciferase. pGB323, pGB324, pGB325, pGB326, pGB327, pGB328, pGB329, pGB346, and pGB347 were made by PCR-amplifying (i) from an *E. coli* MG1655 colony using primer JW9288 and each of JW9989, JW10122, JW10144, JW10145, JW10146, JW10147, JW10148, JW101848, and JW10849, respectively, and (ii) from geneBlock GB007 (*Supplementary file 1F*) using primers JW10123+JW9965. Pairs of PCR products were combined using splicing by overlap extension (*Horton et al., 1990*), and the resultant products were cloned into pGB135 cut with *Bam*HI and *Eco*RI. pGB308 is equivalent to these plasmids except *toiL* coding

sequence was replaced with the beginning of the *ermCL* sequence (codons 1–9). pGB308 was made by PCR-amplifying (i) from an *E. coli* MG1655 colony using primer JW9288+JW9989, and (ii) from geneBlock GB007 (*Supplementary file 1F*) using primers JW9990+JW9965. Pairs of PCR products were combined using splicing by overlap extension (*Horton et al., 1990*), and the resultant products were cloned into pGB135 cut with *Bam*HI and *Eco*RI.

Plasmids pGB322 and pGB318 are derivatives of pPro24 (*Lee and Keasling, 2005*), which contains a propionate-inducible promoter. pGB322 and pGB318 include a complete rRNA operon, either wild-type or mutant (ΔT1917 23S rRNA), respectively. The rRNA operons in both plasmids are fusions of the *rrnB* (upstream of 23S position + 1917) and *rrnC* (position + 1917 and downstream) operons. pGB318 was made first by PCR-amplifying from an *E. coli* MG1655 colony using primers JW10036+JW9737 and JW9740+JW10037, combining the two PCR products using splicing by overlap extension (*Horton et al., 1990*), and cloning the resultant product into pPro24 cut with *Nhe*I and *Bam*HI. pGB322 was made by PCR-amplifying from pGB318 using primers JW10036+JW10104 and JW9740+JW10105, combining the two PCR products using splicing by overlap extension (*Horton et al., 1990*), and cloning the resultant product into pPro24 cut with *Nhe*I and *Bam*HI.

Plasmid pGB72 is a derivative of pET-PesaRlux (*Shong and Collins, 2013*) that has a constitutive promoter *Burr et al., 2000* followed by 568 bp of a reverse-complemented *Hin*dIII fragment from the *rrsB* gene that is known to function as a Rho terminator (*Li et al., 1984*). pGB72 was made by annealing and extending primers JW7977b+7979b, PCR-amplifying from plasmid pSL103 (*Li et al., 1984*) using primers JW7994 and J7995, combining the two PCR products using splicing by overlap extension (*Horton et al., 1990*), and cloning the resultant product into pET-PesaRlux cut with *Xho*I and *Bam*HI.

## Isolation and identification of *trans*-acting mutants

The *trans*-acting mutant genetic selection was performed as described previously (*Baniulyte et al., 2017*). Briefly, cultures of MG1655 Δ*lacZ* with pJTW100 were grown at 37°C in LB medium. 100 μl of overnight culture was washed and plated on M9 + 0.2% lactose agar. Spontaneous survivors were tested for increased plasmid copy number or *cis*-acting mutations near *topAI*; these mutants were eliminated. Chromosomal mutations were identified either by PCR-amplification and sequencing of *rho* or by whole-genome sequencing. Rho mutants identified this way included F62S, F62V, G63V (isolated three times), R66C, R66S, Y80C, Y80D, Y80N, Y80S, S82F (isolated twice), and P138L. Rho mutants were also detected by transducing a wild-type *rho* locus and looking for phenotypic reversion, or by introducing and assaying a Rho-dependent termination luciferase reporter plasmid (pGB72). Total genomic DNA was extracted from the three remaining mutants following the manufacturer's protocol (QIAGEN, catalog # 69504). SNPs were identified using whole-genome sequencing, as described previously (*Singh et al., 2016*).

## Quantification of RNAP association with DNA by chromatin immunoprecipitation coupled with real-time PCR (ChIP-qPCR)

20 ml *E. coli* wild-type (MG1655) or *rho* mutant (CRB016) cells were grown in LB at 37°C with shaking. When the cultures reached an $OD_{600}$ of ~1.0, tetracycline (Sigma #T3383) was added to one set of samples at a final concentration of 0.5 μg/ml. All untreated and tetracycline-treated samples (three biological replicates) were left at 37°C with shaking for 17 min, after which 500 μl of each sample was pelleted and frozen for total RNA extraction later. For ChIP, the remainder of each sample was cross-linked with 1% formaldehyde (final concentration) and processed as previously described (*Stringer et al., 2014*). For immunoprecipitation, 800 μl of crosslinked chromatin was incubated with 25 μl Protein A Sepharose slurry (50%) in TBS and 1 μl of antibody against RNAP β (BioLegend, #663003, clone: NT63) for 90 min at room temperature while rotating. RNAP enrichment was normalized using the $2^{-\Delta\Delta Ct}$ method to the corresponding 'input' control sample and either the *rhoL* region (for measurements within *rho*) or the *topAI* promoter region (for measurements within *topAI-yjhQP*), as previously described (*Bastet et al., 2017*). All real-time PCR samples were set up in technical duplicates.

## Quantification of RNA levels by reverse transcription coupled with real-time PCR (RT-qPCR)

Total RNA was extracted from frozen pellets (see above) using a Quick-RNA Miniprep Kit (Zymo Research, #R1054) following the manufacturer's protocol. Approximately 136 ng of total RNA for each sample was reverse-transcribed using MultiScribe Reverse Transcriptase (Invitrogen, #4311235) following the manufacturer's protocol. A 'no reverse transcriptase' control was also included to assess DNA contamination. The resulting cDNA was diluted 1:3 and used as a template for quantitative real-time PCR. RNA expression values were obtained using the $2^{-\Delta\Delta Ct}$ method, normalizing to a region within the *mreB* gene. Values were further normalized to those in wild-type cells (MG1655) without tetracycline treatment. All real-time PCR samples were set up in technical duplicates.

## RNA structural prediction using sequence identity among homologues

We used PETfold (*Seemann et al., 2011*; *Seemann et al., 2008*) to predict the *topAI* UR RNA secondary structure based on a sequence alignment of selected homologous sequences (*Figure 6—figure supplement 1A*). All sequences are from species with at least 50% TopAI amino acid identity to the *E. coli* K-12 protein.

## SHAPE-seq

A total of three biological replicates of the strain GB001+pJTW100 were grown to an $OD_{600}$ of ~1.2 in LB medium at 37°C. Cultures were split, and one set of two replicates was treated with 0.2 µg/ml tetracycline (T3383, Sigma) for 90 min. Cultures were subjected to the in-cell SHAPE-seq procedure previously described for poorly expressed mRNAs (*Watters et al., 2016b*). The SHAPE reagent 1M7 was purchased from MedChem Express (HY-D0913). SHAPE-seq libraries were sequenced using an Illumina MiSeq instrument (251 nt, paired-end reads). Sequencing data were analyzed using the *spats* pipeline (*Watters et al., 2016a*). Raw reactivities are listed in *Supplementary file 1C*. Significant changes in reactivity and corresponding adjusted p-values upon tetracycline treatment were determined using the DESeq2 R package (*Love et al., 2014*), comparing untreated and tetracycline-treated sample reactivities at each position using three ('Untreated') and two ('Treated') biological replicates from each sample. The mRNA structure of the *topAI* UR (*Figure 6A*) was predicted using mFOLD (*Zuker, 2003*) and drawn using StructureEditor (https://rna.urmc.rochester.edu/GUI/html/StructureEditor.html).

## Reporter assays

Bacterial cultures for luciferase or β-galactosidase assays were grown at 37°C in LB medium to an $OD_{600}$ of 0.5–0.6 unless otherwise indicated in the figure legend. Cultures were grown in 15 ml round-bottom tubes except for antibiotic titration assays (*Supplementary file 1B*; see below). Tetracycline (Sigma #T3383), spectinomycin (Sigma-Aldrich #S4014), erythromycin (Sigma-Aldrich #E5389), retapamulin (Sigma-Aldrich #CDS023386), tylosin (Sigma-Aldrich #T6134), chloramphenicol (Sigma-Aldrich #C0378), kasugamycin, gentamicin (Sigma-Aldrich #G1272), amikacin, streptomycin, apramycin, or hygromycin (Millipore #400052) antibiotics were added at indicated concentrations and timepoints. Reported epidemiological cut-off values for antibiotics reported by EUCAST (*Kahlmeter and Turnidge, 2022*) are tetracycline, 8 µg/ml; spectinomycin, 64 µg/ml; chloramphenicol, 16 µg/ml; gentamicin, 2 µg/ml; amikacin, 8 µg/ml; streptomycin, 16 µg/ml. Minimum inhibitory concentrations (MICs) have been reported for tylosin, 512 µg/ml (*Andersen et al., 2012*) kasugamycin, 500 µg/ml (*Lange et al., 2017*) erythromycin, 150 µg/ml (*Suvorov et al., 1988*) apramycin, 32 µg/ml (*Yang et al., 2020*) retapamulin, 12.5 µg/ml (*Meydan et al., 2019*) and hygromycin, 150 µg/ml (*McGaha and Champney, 2007*). These values were used as a guide for selecting antibiotic concentrations to test in reporter assays. MICs can vary considerably between strains, so the MICs and epidemiological cut-off values listed above may differ from the actual values for *E. coli* K-12.

Plasmid-encoded wild-type and mutant rRNA overexpression (*Figure 2B*) was induced by adding 12 mM sodium propionate (Sigma-Aldrich #P1880) to bacterial cultures at an $OD_{600}$ of ~0.2. β-galactosidase assays were performed as previously described (*Baniulyte et al., 2017*). For luciferase assays, each cell culture (200 µl) was aliquoted into a 96-well plate with four technical replicates each. Luminescence readings for were taken using a Biotek Synergy 2 instrument. Luminescence counts (RLU) were adjusted for $OD_{600}$ and reported as $RLU/OD_{600}$. Antibiotic concentrations and lengths of

treatments are indicated in the figure legends. Antibiotic titration assays (*Supplementary file 1B*) were performed in 96-well deep-well plates. Briefly, 1.5 ml of GB001 cells containing the pGB202 plasmid was grown to an $OD_{600}$ of ~1.0, aliquoted into a 96-well deep-well plate, and treated with antibiotics at the indicated concentrations for 90 min. 100 µl of each bacterial culture was used for luminescence detection, as described above.

### Ribosome profiling (Ribo-seq)

200 ml of *E. coli* MG1655 cells were grown in LB ±1 µg/ml tetracycline to an $OD_{600}$ of ~1.1. Filtering was started at 17 min post-treatment. Monosomes were isolated and libraries were constructed as previously described (*Smith et al., 2022*), with minor modifications: the 3'-linked oligonucleotide ligation reaction was carried out overnight at 16°C; circularized, reverse-transcribed cDNA was PCR-amplified for 6–8 cycles using universal primer JW8835 and sample-specific primers JW3249 (no antibiotic) or JW10755 (tetracycline).

Determination of Ribo-seq sequence read coverage was performed as described previously (*Smith et al., 2022*). Briefly, sequence reads were trimmed using a custom Python script to remove the ligated adaptor sequence (code available at https://github.com/wade-lab/Mtb_Ribo-RET; *Wade, 2021*) and mapped to reference genome MG1655 NC_000913.3 using Rockhopper (*McClure et al., 2013*). Sequence read coverage for sequence 3' ends (presumed to correspond to the positions of the downstream edge of ribosomes) was determined for the whole genome, normalizing to total read count (reads per million; RPM). The metagene plot showing sequence read coverage around start codons (*Figure 7C*) was generated as described previously (*Smith et al., 2022*). Normalized Ribo-seq coverage for all annotated ORFs was determined using differential expression analysis with Rockhopper (*McClure et al., 2013*), excluding the first and last 30 nt of each gene.

### Acknowledgements

We thank Randall Morse, Pan Li, and Allen Buskirk for helpful comments on the manuscript. We thank the Wadsworth Center Applied Genomic Technologies Core Facility for DNA sequencing. We thank the Wadsworth Center Tissue Culture and Media Core Facility and Glassware Facility for technical support. This work was supported in part by the NIH Director's New Innovator Award Program, 1DP2OD007188 (JTW) and National Institutes of Health grant 1R35GM144328 (JTW).

## Additional information

### Funding

| Funder | Grant reference number | Author |
| --- | --- | --- |
| National Institutes of Health | 1DP2OD007188 | Joseph T Wade |
| National Institute of General Medical Sciences | 1R35GM144328 | Joseph T Wade |

The funders had no role in study design, data collection and interpretation, or the decision to submit the work for publication.

### Author contributions

Gabriele Baniulyte, Conceptualization, Software, Formal analysis, Investigation, Methodology, Writing – original draft, Writing – review and editing; Joseph T Wade, Conceptualization, Formal analysis, Supervision, Funding acquisition, Writing – original draft, Project administration, Writing – review and editing

### Author ORCIDs

Gabriele Baniulyte (ID) https://orcid.org/0000-0003-0235-7938
Joseph T Wade (ID) https://orcid.org/0000-0002-9779-3160

Reviewer #1 (Public review): https://doi.org/10.7554/eLife.101217.4.sa1
Reviewer #2 (Public review): https://doi.org/10.7554/eLife.101217.4.sa2
Reviewer #3 (Public review): https://doi.org/10.7554/eLife.101217.4.sa3
Author response https://doi.org/10.7554/eLife.101217.4.sa4

## Additional files

### Supplementary files
MDAR checklist

Supplementary file 1. Supplementary tables.

### Data availability
Raw Illumina sequencing data are available from EBI ArrayExpress using accession numbers E-MTAB-8365 (SHAPE-seq) and E-MTAB-12242 (Ribo-seq).

The following datasets were generated:

| Author(s) | Year | Dataset title | Dataset URL | Database and Identifier |
|---|---|---|---|---|
| Baniulyte G, Smith C, Wade J | 2019 | An antibiotic-sensing leader peptide regulates translation and premature Rho-dependent transcription termination of the topAI gene in *Escherichia coli* | https://www.ebi.ac.uk/biostudies/ArrayExpress/studies/E-MTAB-8365?query=E-MTAB-8365 | EBI ArrayExpress, E-MTAB-8365 |
| Stringer A | 2022 | Ribo-seq in *Escherichia coli* MG1655 treated withTetracycline and Tylosin | https://www.ebi.ac.uk/biostudies/ArrayExpress/studies/E-MTAB-12242?query=E-MTAB-12242 | EBI ArrayExpress, E-MTAB-12242 |

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
