## [Editor Report · eLife Assessment]

In this **important** study, Baniulyte and Wade provide **convincing** evidence that translation of a short ORF denoted *toiL* positioned upstream of the *topAI-yjhQP* operon is responsive to different ribosome-targeting antibiotics, consequently controlling translation of the TopAI toxin as well as Rho-dependent transcription termination. Strengths of the study include combining a genetic screen to identify 23S rRNA mutations that affect *topA1* expression and a creative approach to map the different locations of ribosome stalling within *toiL* induced by different antibiotics, with ribosome profiling and RNA structure probing by SHAPE to examine consequences of different antibiotics on *toiL*-mediated regulation. The work leaves unanswered how bacteria benefit by activating expression of the genes using the proposed strategy and the mechanism underlying ToiL's sensing of structurally distinct antibiotics.

---

## [Referee Report · Reviewer #1 (Public review)]

Summary:

The manuscript reports that expression of the *E. coli* operon topAI/yjhQ/yjhP is controlled by the translation status of a small open reading frame, that authors have discovered and named toiL, located in the leader region upstream of the operon. Authors propose the following model for topAI activation: Under normal conditions, toiL is translated but topAI is not expressed because of Rho-dependent transcription termination within the topAI ORF and because its ribosome binding site and start codon are trapped in an mRNA hairpin. Ribosome stalling at various codons of the toiL ORF, prompted in this work by some ribosome-targeting antibiotics, triggers an mRNA conformational switch which allows translation of topAI and, in addition, activation of the operon's transcription because presence of translating ribosomes at the topAI ORF blocks Rho from terminating transcription. The model is appealing and several of the experimental data mainly support it. However, it remains unanswered what is the true trigger of the translation arrest at toiL and what is the physiological role of the induced expression of the topAI/yjhQ/yjhP operon.

---

## [Referee Report · Reviewer #2 (Public review)]

Summary:

Baniulyte and Wade describe how translation of an 8-codon uORF denoted toiL upstream of the topAI-yjhQP operon is responsive to different ribosome-targeting antibiotics, consequently controlling translation of the TopAI toxin as well as Rho-dependent termination with the gene.

Strengths:

The authors used multiple different approaches such as a genetic screen to identify factors such as 23S rRNA mutations that affect topA1 expression and ribosome profiling to examine the consequences of various antibiotics on toiL-mediated regulation.

Weaknesses: Future experiments will be needed to better understand the physiological role of the toiL-mediated regulation and elucidate the mechanism of specific antibiotic sensing.

The results are clearly described, and the revisions have helped to improve the presentation of the data.

---

## [Referee Report · Reviewer #3 (Public review)]

The authors provide convincing data to support an elegant model in which ribosome stalling by ToiL promotes downstream topAI translation and prevents premature Rho-dependent transcription termination. However, the physiological consequences of activating topAI-yjhQP expression upon exposure to various ribosome-targeting antibiotics remain unresolved. The authors have satisfactorily addressed all major concerns raised by the reviewers, particularly regarding the SHAPE-seq data. Overall, this study underscores the diversity of regulatory ribosome-stalling peptides in nature, highlighting ToiL's uniqueness in sensing multiple antibiotics and offering significant insights into bacterial gene regulation coordinated by transcription and translation.

[Editors' note: The earlier public reviews are included. No additional reviews were requested.]

---

## [Author Response]

The following is the authors’ response to the previous reviews

**Public Reviews:**

**Reviewer #1 (Public review):**
Summary:The manuscript reports that expression of the *E. coli* operon topAI/yjhQ/yjhP is controlled by the translation status of a small open reading frame, that authors have discovered and named toiL, located in the leader region upstream of the operon. Authors propose the following model for topAI activation: Under normal conditions, toiL is translated but topAI is not expressed because of Rho-dependent transcription termination within the topAI ORF and because its ribosome binding site and start codon are trapped in an mRNA hairpin. Ribosome stalling at various codons of the toiL ORF, prompted in this work by some ribosome-targeting antibiotics, triggers an mRNA conformational switch which allows translation of topAI and, in addition, activation of the operon's transcription because presence of translating ribosomes at the topAI ORF blocks Rho from terminating transcription. The model is appealing and several of the experimental data mainly support it. However, it remains unanswered what is the true trigger of the translation arrest at toiL and what is the physiological role of the induced expression of the topAI/yjhQ/yjhP operon.
**Reviewer #2 (Public review):**
Summary:Baniulyte and Wade describe how translation of an 8-codon uORF denoted toiL upstream of the topAI-yjhQP operon is responsive to different ribosome-targeting antibiotics, consequently controlling translation of the TopAI toxin as well as Rho-dependent termination with the gene.Strengths:The authors used multiple different approaches such as a genetic screen to identify factors such as 23S rRNA mutations that affect topA1 expression and ribosome profiling to examine the consequences of various antibiotics on toiL-mediated regulation.Weaknesses:Future experiments will be needed to better understand the physiological role of the toiL-mediated regulation and elucidate the mechanism of specific antibiotic sensing.The results are clearly described, and the revisions have helped to improve the presentation of the data.
**Reviewer #3 (Public review):**
In this revised manuscript, the authors provide convincing data to support an elegant model in which ribosome stalling by ToiL promotes downstream topAI translation and prevents premature Rho-dependent transcription termination. However, the physiological consequences of activating topAI-yjhQP expression upon exposure to various ribosome-targeting antibiotics remain unresolved. The authors have satisfactorily addressed all major concerns raised by the reviewers, particularly regarding the SHAPE-seq data. Overall, this study underscores the diversity of regulatory ribosome-stalling peptides in nature, highlighting ToiL's uniqueness in sensing multiple antibiotics and offering significant insights into bacterial gene regulation coordinated by transcription and translation.
**Recommendations for the authors:**

**Reviewer #1 (Recommendations for the authors):**
- Showing the ribosome density profiles of topAI/yjhQP and toiL in control and tetracycline treated cells is necessary to support that ribosome arrest at toiL increases translation of topAI/yjhQP.

Figure 7B shows ribosome density around the start of *toiL*. Ribosome density increases across *topAI* in the presence of tetracycline, but we have opted not to show this region because we cannot say whether the increase in ribosome occupancy (represented in Figure 7A) is due to an increase in translation efficiency, RNA level, or both.

- The subinhibitory antibiotic concentrations used in the reporter assays were based on MICs reported in the literature. This is not appropriate since MICs can greatly vary between strains, antibiotic solution stocks, and experimental conditions.

Reported MICs were used as an initial guide for selecting antibiotic concentrations to test in our reporter assays. We have added text to indicate this, and to highlight that MICs vary considerably between strains.

- toiL sequence may have evolved to maintain base-pairing with the topAI upstream region rather than, as authors suggest in Discussion, to respond to antibiotic-mediated arrest in an amino acid sequence specific manner.

We have chosen to frame this as speculation.

- Authors may consider commenting on the possibility that chloramphenicol does not induce because ToiL lacks alanine residues, whose presence at specific places of a nascent protein have been shown to promote chloramphenicol action (2016 PNAS 113:12150; 2022 NSMB 29:152).

This is a great point as none of our stalling reporters included an ORF with alanine. We now include a short paragraph in the Discussion section to raise this possibility.

- Tetracycline was added at the "subinhibitory concentration" of 8 ug/mL for the reporter assays but at 1 ug/mL for the ribosome profiling experiments. Authors should explain what was the rational for this.

We think the reviewer is mixing up the epidemiological cut-off value of 8 ug/mL with the concentration used in experiments (0.5-1 ug/mL for reporter assays and ribosome profiling). The text was confusing, so we have added a sentence to the Methods section to indicate that epidemiological cut-off values and MICs were only a guide for selecting antibiotic concentrations to test.

**Reviewer #2 (Recommendations for the authors):**
I wish the authors had been slightly less dismissive of the reviewers' comments. At a minimum, it would be nice if the authors could be consistent about the ribosome representation throughout the manuscript;

We apologize if our previous responses gave the impression of being dismissive. That was certainly not our intention. We greatly value the reviewers' feedback, and we appreciate the opportunity to clarify any misunderstandings. We believe the reviewer is referring to the different shape and color of the ribosome in Figures 8 and 9, and Figure 8 figure supplement 2, which we have now corrected.